# Demystifying the Token Dynamics of Deep Selective State Space Models

**Thieu N. Vo**
Department of Mathematics
National University of Singapore
thieuvo@nus.edu.sg

**Duy-Tung Pham**
FPT Software AI Center
tungpd10@fpt.com

**Xin T. Tong**\*
Department of Mathematics
National University of Singapore
mattxin@nus.edu.sg

**Tan M. Nguyen**\*
Department of Mathematics
National University of Singapore
tanmn@nus.edu.sg

## Abstract

Selective state space models (SSM), such as Mamba, have gained prominence for their effectiveness in modeling sequential data. Despite their outstanding empirical performance, a comprehensive theoretical understanding of deep selective SSM remains elusive, hindering their further development and adoption for applications that need high fidelity. In this paper, we investigate the dynamical properties of tokens in a pre-trained Mamba model. In particular, we derive the dynamical system governing the continuous-time limit of the Mamba model and characterize the asymptotic behavior of its solutions. In the one-dimensional case, we prove that only one of the following two scenarios happens: either all tokens converge to zero, or all tokens diverge to infinity. We provide criteria based on model parameters to determine when each scenario occurs. For the convergent scenario, we empirically verify that this scenario negatively impacts the model's performance. For the divergent scenario, we prove that different tokens will diverge to infinity at different rates, thereby contributing unequally to the updates during model training. Based on these investigations, we propose two refinements for the model: excluding the convergent scenario and reordering tokens based on their importance scores, both aimed at improving practical performance. Our experimental results validate these refinements, offering insights into enhancing Mamba's effectiveness in real-world applications. The code is publicity available at https://github.com/Fsoft-AIC/Mamba-token-dynamic.

## 1 Introduction

State space models (SSMs) have undergone significant advancements to mitigate the computational inefficiency associated with the sequence modeling (Gu et al., 2022b; 2021; 2022a; Gupta et al., 2022; Li et al., 2023; Kosma et al., 2023; Orvieto et al., 2023; Smith et al., 2023), and they have been successfully applied in various domains involving continuous signal data such as audio and vision (Goel et al., 2022; Nguyen et al., 2022a; Saon et al., 2023). SSMs can be viewed as a combination of recurrent neural networks (RNNs) and convolutional neural networks (CNNs), drawing on concepts from traditional state space frameworks (Kalman, 1960). Unlike transformers, which experience quadratic scaling with respect to input sequence length and exhibit significant computational demands (Vaswani et al., 2017), SSMs are designed for efficient computation through recurrence or convolution, achieving linear or near-linear scaling as the sequence length increases. Furthermore, SSMs incorporate robust mechanisms for capturing long-range dependencies (Gu et al., 2020) across various data types, and they have excelled in benchmark tasks such as the Long Range Arena (Tay et al., 2021).

---

\* Co-last authors. Please correspond to: thieuvo@nus.edu.sg and tanmn@nus.edu.sg

Selective SSMs such as Mamba is a particular SSM which involves the selective state-space layer (S6) as their core component (Gu & Dao, 2024). In an S6 layer, parameters are functions of the input, endowing the SSM with the content awareness ability. Mamba has showcased exceptional performance across a variety of applications, such as language modeling (Pióro et al., 2024; Lenz et al., 2025), image processing (Liu & Li, 2024; Zhu et al., 2024), video analysis (Yang et al., 2024; Li et al., 2024a), medical imaging (Ma & Wang, 2024; Wang et al., 2024b), tabular data analysis (Ahamed & Cheng, 2024), reinforcement learning (Ota, 2024), point-cloud analysis (Liang et al., 2025), graph processing (Wang et al., 2024a), and $N$-dimensional sequence modeling (Li et al., 2024b). It has been argued that Mamba's significant success across various domains stems from its ability to compute complicated representations for different data types (Rezaei Jafari et al., 2025; Patro & Agneeswaran, 2024; Xu et al., 2024; Zhu et al., 2024).

Despite their remarkable empirical success, a thorough theoretical understanding of deep selective SSM is still lacking, which poses challenges for their advancement and application in high-fidelity tasks. As these models are scaled at a remarkable rate, understanding their internal mechanisms has become a major problem. In this paper, we take an initial step toward addressing this gap by analyzing the dynamical properties of tokens in a pretrained Mamba.

## 1.1 BACKGROUND: MAMBA

The building block of a selective state-space model is a Mamba block, which is built on top of the S6 layers, along with linear layers, convolutions, and other token-wise operators. The S6 layer, which serves as the core component of the Mamba block, plays a pivotal role in the success of these models.

Formally, an S6 layer is defined as a function that maps a sequence of tokens $\mathbf{x} = (x_1, \ldots, x_L) \in \mathbb{R}^{D \times L}$ to another sequence of tokens $\mathbf{y} = (y_1, \ldots, y_L) \in \mathbb{R}^{D \times L}$ (with the same number of channels). In this context, each vector $x_l$ (or $y_l$) represents a token, while the whole sequence $\mathbf{x} = (x_1, \ldots, x_L)$ (or $\mathbf{y} = (y_1, \ldots, y_L)$) is called a prompt. For each $d = 1, \ldots, D$, the $d$-th channel output $\hat{y}_d = (y_{d1}, \ldots, y_{dL}) \in \mathbb{R}^L$ is determined recurrently from the corresponding $d$-th channel input $\hat{x}_d = (x_{d1}, \ldots, x_{dL}) \in \mathbb{R}^L$ via a sequence of hidden states $h_{d1}, \ldots, h_{dL} \in \mathbb{R}^N$ as follows:

$$\begin{cases} h_{dl} = \overline{A}_{dl} \cdot h_{d,l-1} + \overline{B}_{dl} \cdot x_{dl}, & h_{d0} = 0, \\ y_{dl} = C_l \cdot h_{dl}, \end{cases} \tag{1}$$

for $l = 1, \ldots, L$. Here, the matrices $\overline{A}_{dl}, \overline{B}_{dl}$, and $C_l$ are input-dependent and time-varying, determined by

$$\overline{A}_{dl} = e^{\Delta_d(x_l)A_d}, \quad \overline{B}_{dl} = \Delta_d(x_l)S_B \cdot x_l, \quad C_l = (S_C \cdot x_l)^\top, \tag{2}$$

where $\Delta_d : \mathbb{R}^D \to \mathbb{R}$ is the step size function at the $d$-th channel and is defined by

$$\Delta_d(u) = \text{softplus}(S_{\Delta,d}u) = \ln\left(1 + e^{S_{\Delta,d} \cdot u}\right), \qquad u \in \mathbb{R}^D. \tag{3}$$

The hidden matrices $A_d \in \mathbb{R}^{N \times N}$ and the step size vectors $S_{\Delta,d} \in \mathbb{R}^{1 \times D}$ for $d = 1, \ldots, D$, as well as the input and output matrices $S_B, S_C \in \mathbb{R}^{N \times D}$, are learnable from input data. The matrices $A_d$ are diagonal matrices with negative eigenvalues. In recent developments, the matrices $A_d$ are chosen to be of the scalar form $A_d = -a_d I_N$ for some positive number $a_d$ (see (Dao & Gu, 2024)), which does not compromise model performance. We will also use this scalar form for $A_d$ in our paper. In our context, the matrix $S_C^\top S_B \in \mathbb{R}^{D \times D}$, which we will refer to as the *input-output matrix*, plays an essential role in characterizing the dynamical properties of tokens.

## 1.2 CONTRIBUTION

This paper aims to describe the dynamical properties of tokens in a pre-trained Mamba model. To achieve this, we consider the dynamical system governing the continuous-time limit of Mamba. The dynamical properties of tokens are described through the asymptotic behavior of the solutions of the dynamical system. Additionally, we empirically investigate the relationship between the dynamical properties of the tokens and model performance. In summary, our main contributions are three-fold:

1. In the one-dimensional case, we prove that only one of the following two scenarios occurs: either all tokens converge to zero, or all tokens diverge to infinity. We provide criteria based on model parameters to determine when each scenario occurs. Our experiments suggest that these observations generally hold in high-dimensional cases.

2. For the convergent scenario, we empirically verify that this situation negatively impacts the model's performance. In contrast, for the divergent scenario, we prove that different tokens will diverge to infinity at different rates, thereby contributing unequally to the updates during model training.

3. Based on these investigations, we propose two refinements for the model: (i) we exclude the convergent scenario before training as it negatively impacts performance, and (ii) we reorder the tokens according to their ascending importance scores, as different tokens are prioritized unequally during training.

We empirically demonstrate the benefits of our token rendering method in improving the model's accuracy and convergence speed compared to the baseline Mamba on the large-scale ImageNet classification task (Deng et al., 2009)

**Organization of the paper**    After surveying related works in Section 2, we introduce the dynamical system that governs the continuous-time limit of Mamba. In Section 4, we characterize the dynamical properties of tokens based on model parameters and determine the divergence rate in the divergence scenario. Based on the findings in Section 4, we propose two refinements for the model in Section 5: excluding unfavorable scenarios and reordering tokens before training. We conclude the paper with a discussion of conclusions and limitations in Section 6.

**Notations.**    We use small bold letters (e.g., $\mathbf{x}, \mathbf{y}, \mathbf{u}$) to denote sequences of tokens. Tokens can be either vectors or scalars, depending on the context, and are denoted by small normal letters (e.g., $x_i, y_j, u_k$), where the indices emphasize their position in the sequence. We use capital letters (e.g., $X, Y, P, A, B, C$) to represent matrices.

## 2    RELATED WORK

Several studies have analyzed the expressivity and generalization of Mamba from a theoretical perspective. For instance, the authors in (Ali et al., 2024) demonstrated that the S6 layer can be interpreted as a variant of softmax-free attention with linear complexity, subsequently proving that Mamba is more expressive than transformers. Conversely, (Muca Cirone et al., 2025) employed tools from Rough Path Theory to show that diagonal selective state space models (SSMs), such as Mamba, possess less expressive power than their non-diagonal counterparts. Furthermore, (Merrill et al., 2024) utilized circuit complexity theory to establish that both SSM variants and transformers share the same expressive power, as they belong to the complexity class $TC^0$, which can be decided by polynomial-sized Boolean circuits. In contrast, the authors in (Jelassi et al., 2024) provided both theoretical and empirical evidence that transformers surpass state space models in their copying capabilities. All of these works aimed to provide a theoretical framework to understand the expressivity and generalization of Mamba.

In this paper, we examine the dynamical properties of tokens in a pretrained Mamba model. Following the methodology outlined in (Geshkovski et al., 2023; 2024), we define an idealized model of Mamba by interpreting the discrete layer indices as continuous time variables. This idealized perspective was used in ResNets (Chen et al., 2018; Haber & Ruthotto, 2017; Abdullaev & Nguyen, 2025) and neural ODEs (Lin & Jegelka, 2018; Zhang et al., 2020; Xia et al., 2021; Li et al., 2022; Tabuada & Gharesifard, 2022; Nguyen et al., 2022b; Ruiz-Balet & Zuazua, 2023; Cheng et al., 2025). Our model focuses on the S6 layer, which is central to Mamba and contributes to its success across various domains.

## 3    SELECTIVE STATE SPACE DYNAMICS

In this section, we introduce the dynamical system governing the continuous-time counterpart of Mamba. This dynamical system will provide an effective tool to characterize the dynamical properties of tokens in Mamba as well as the impact of these properties on model performance in the next section.

### 3.1    BACKGROUND: CONTINUOUS-TIME LIMIT OF A DEEP NEURAL NETWORK

To ensure a clear presentation of our results, we draw upon the literature concerning the dynamical systems that govern the continuous-time limit of deep neural networks (DNNs). Generally speaking, data in a DNN is processed sequentially, layer by layer, resulting in a discrete-time dynamical system

(LeCun et al., 2015). A notable example is residual neural networks (ResNets) and their continuous-time counterparts known as neural ODEs (Chen et al., 2018; Haber & Ruthotto, 2017; Nguyen et al., 2020; Wang et al., 2022). Each layer of ResNet is a residual block, transforms an input vector $x \in \mathbb{R}^D$ to an output vector $z \in \mathbb{R}^D$ via a two-layer feed-forward neural network $x \mapsto y(x, \theta)$, parametrized by $\theta$, and a skip-connection as:

$$z = x + y(x, \theta).$$

The continuous-time counterparts of ResNet is conceptualized as a flow map that inputs a vector $x(0) \in \mathbb{R}^D$ and outputs another vector $x(T) \in \mathbb{R}^D$, which is processed via the corresponding dynamical system

$$x'(t) = y(x(t), \theta(t)), \quad t \in (0, T).$$

There exists a body of work investigating the interpolation, approximation, and controllability properties of these DNN architectures (Lin & Jegelka, 2018; Zhang et al., 2020; Li et al., 2022; Tabuada & Gharesifard, 2022; Ruiz-Balet & Zuazua, 2023; Cheng et al., 2025; Nguyen et al., 2024).

### 3.2 CONTINUOUS-TIME LIMIT OF MAMBA

Unlike ResNets and neural ODEs, Mamba represents a function on a sequence of $D$-dimensional tokens rather than solely on an individual input, as discussed in Section 1.1. In order to derive the dynamical system governing the continuous-time counterpart of Mamba, we eliminate the hidden states $h_{dl}$ in system (1) to obtain the following form (see (Ali et al., 2024)):

$$y_{dl} = \sum_{j=1}^{l} P_{dlj} x_{dj}, \tag{4}$$

or equivalently, $\hat{y}_d = P_d \hat{x}_d$, where

$$P_d = \begin{bmatrix} P_{d11} & 0 & \dots & 0 \\ P_{d21} & P_{d22} & \dots & 0 \\ \vdots & \vdots & \ddots & \vdots \\ P_{dL1} & P_{dL2} & \dots & P_{dLL} \end{bmatrix}, \tag{5}$$

and

$$P_{dlj} = \begin{cases} x_l^\top \left(S_C^\top S_B\right) x_l \cdot \Delta_d(x_l), & \text{if } l = j, \\ x_l^\top \left(S_C^\top S_B\right) x_j \cdot \Delta_d(x_j) \cdot \exp\left(-a_d \sum_{k=j+1}^{l} \Delta_d(x_k)\right), & \text{if } l > j. \end{cases} \tag{6}$$

In the above formulation, the upper triangular matrices $P_d$ ($d = 1, \dots, D$) represent the hidden attention scores between the $d$-th channel input and output. Following the setting of (Ali et al., 2024), we call $P = [P_1, \dots, P_D]$ the hidden attention tensor of the S6 layer.

Following the common approaches used in studying the continuous-time counterparts of ResNets (Chen et al., 2018; Haber & Ruthotto, 2017) and Transformers (Geshkovski et al., 2023; 2024), we consider the layer index of a deep selective state space model as a time variable and interpret the selective state space model as the discrete-time version of a parametrized dynamical system of the form

$$\frac{d}{dt} x_{dl}(t) = \sum_{j=1}^{l} P_{dlj}(t) x_{dj}(t), \quad t \in [0, +\infty), \tag{7}$$

$$x_{dl}(0) = x_{dl0} \in \mathbb{R},$$

for $l = 1, \dots, L$ and $d = 1, \dots, D$, where

$$P_{dlj}(t) = \begin{cases} x_l(t)^\top \left(S_C^\top S_B\right) x_l(t) \cdot \Delta_d(x_l(t)), & \text{if } l = j, \\ x_l(t)^\top \left(S_C^\top S_B\right) x_j(t) \cdot \Delta_d(x_j(t)) \cdot \exp\left(-a_d \sum_{k=j+1}^{l} \Delta_d(x_k(t))\right), & \text{if } l > j, \end{cases} \tag{8}$$

and $\Delta_d(u) = \text{softplus}(S_{\Delta, d} u)$. It is important to mention that the time $t$ in this dynamical system represents the depth direction of Mamba model.

Similar to (Geshkovski et al., 2023; 2024), we have focused exclusively on S6 layer, which is the key component of Mamba, and the skip-connection in the above dynamical system. We have ignored other token-wise operators such as layer normalization, convolution, and linear functions. In addition, we assume the parameters $A_d, S_{\Delta,d}, S_B, S_C$ are time-independent. These assumptions are primarily motivated by mathematical convenience. Nevertheless, such a weight-sharing mechanism is used in practice to reduce the number of trainable parameters, as seen, for example, in ALBERT (Lan et al., 2020), a transformer-based large language model. The dynamical properties of tokens in deep selective state space models with these additional layers will be discussed in future works.

## 4 DYNAMICAL PROPERTIES OF TOKENS IN MAMBA

In this section, we study the asymptotic behavior of the solution $\mathbf{x}(t)$ of the dynamical system (7), as well as its corresponding attention score $P(t) = (P_{dlj}(t))_{dlj}$ defined in equation (8). This information encodes the dynamical properties of tokens and hidden attention scores in deep selective state space models.

### 4.1 CONVERGENCE AND DIVERGENCE SCENARIOS OF MAMBA'S DYNAMICS

In the standard form of S6, each $d$-th channel output is determined from the corresponding $d$-th channel input using time-variant input-dependent parameters. Therefore, we will consider the single channel dimension case, i.e. $D = 1$, in our setting to avoid overly complicated technicalities. In this case, the input sequence $\mathbf{x} = (x_1, \ldots, x_L) \in \mathbb{R}^L$ is a list of $L$ real numbers. We drop the index $d$ from Eq. (7) and rewrite it as

$$\frac{d}{dt}x_l(t) = \sum_{j=1}^{l} P_{lj}(t)x_j(t), \quad t \in [0, +\infty), \tag{9}$$

$$x_l(0) = x_{l0} \in \mathbb{R},$$

for $l = 1, \ldots, L$, where

$$\Delta(u) = \text{softplus}(S_\Delta u) = \ln\left(1 + e^{S_\Delta u}\right), \qquad u \in \mathbb{R}, \tag{10}$$

and

$$P_{lj}(t) = \begin{cases} \left(S_C^\top S_B\right) x_l(t)^2 \cdot \Delta(x_l(t)), & \text{if } l = j, \\ \left(S_C^\top S_B\right) x_l(t)x_j(t) \cdot \Delta(x_j(t)) \cdot \exp\left(-a \sum_{k=j+1}^{l} \Delta(x_k(t))\right), & \text{if } l > j. \end{cases} \tag{11}$$

In this case, $S_B, S_C \in \mathbb{R}^{N \times 1}$, thus $S_C^\top S_B \in \mathbb{R}$, and the scalar values $S_\Delta$ and $a \in \mathbb{R}$, with $a > 0$, are learnable from input data.

To avoid triviality, we will always assume that the initial datum $x_l(0) = x_{l0}$ are nonzero for all $l$. The main goal of this section is to investigate the asymptotic behavior of the solution $\mathbf{x}(t) = (x_1(t), \ldots, x_L(t))$ of the dynamic (9) and the corresponding hidden attention matrix

$$\mathbf{P}(t) = \begin{bmatrix} P_{11}(t) & 0 & \ldots & 0 \\ P_{21}(t) & P_{22}(t) & \ldots & 0 \\ \ldots & \ldots & \ldots & \ldots \\ P_{L1}(t) & P_{L2}(t) & \ldots & P_{LL}(t) \end{bmatrix},$$

when $t$ increases. The following three scenarios will be considered separately:

1. Convergence scenario: $S_C^\top S_B < 0$;
2. Slow-divergence scenario: $S_C^\top S_B > 0$ and $S_\Delta x_{l0} < 0$ for all $l$;
3. Fast-divergence scenario: $S_C^\top S_B > 0$ and $S_\Delta x_{l0} > 0$ for some $l$.

We will see that, in the first scenario, all tokens and hidden attention scores quickly converge to zero as $t \to \infty$, at rates of $O(1/\sqrt{t})$ and $O(1/t)$, respectively (Theorem 4.1). In the second scenario, all tokens diverge to infinity slowly at a logarithmic rate, while the hidden attention scores still approach zero as $t \to \infty$. We refer to this as the slow-divergence scenario (Theorem 4.3). In the final scenario, one token diverges to infinity very quickly in finite time, causing a blow-up in the corresponding hidden attention scores. We refer to this as the fast-divergence scenario (Theorem 4.4). Table 1 summarizes the dynamical properties of tokens in this work.

Table 1: Summary of the dynamical properties of this work.

| Parameters | | Scenario | Impact on model | Reference |
|---|---|---|---|---|
| $S_C^\top S_B < 0$ | | Convergence | Negative | Theorem 4.1 |
| $S_C^\top S_B > 0$ | $\forall l, \, S_\Delta x_{l0} < 0$ | Slow-divergence | Positive | Theorem 4.3 |
| | $\exists l, \, S_\Delta x_{l0} > 0$ | Fast-divergence | | Theorem 4.4 |

**Remark 1** (Convergence vs. divergence scenarios and their impact). We collectively refer to the slow-divergence and fast-divergence scenarios as the divergence scenarios. By utilizing model parameters to test the signature of $S_C^\top S_B$, we can distinguish between the convergence scenario and the divergence scenarios. However, to distinguish between the slow-divergence and fast-divergence scenarios, additional information from the input data, specifically $x_{l,0}$, is required. In addition, our experiments suggest that the convergence scenario negatively impacts model performance in practice, while the divergence scenario positively impacts model performance. Therefore, the convergence scenario should be excluded before training (see Subsection 5.1).

**Remark 2** (Unequal contribution of tokens during training). In the slow-divergence scenario, when tokens diverge to infinity at infinity, we further prove that tokens with different initial positions will have different rates of divergence. As a consequence, the contribution of tokens to the updates during model training is unequal. Intuitively speaking, the tokens should be rearranged in such a way that those carrying more important information diverge faster than those carrying less important information. In Subsection 5.2, we suggest a method for reordering the tokens before training by projecting them into a learnable line. Our experimental results show that this ordering method leads to better model performance.

**Remark 3** (Higher dimension). When the dimension of the input tokens is $D \geq 1$, the input-output projection matrix $S_C^\top S_B$ is a square matrix of size $D \times D$, which may have complex eigenvalues. We conjecture that the dynamic behavior of the tokens in system (7) can be determined by analyzing the matrix

$$\mu = \frac{1}{2} \left( S_C^\top S_B + (S_C^\top S_B)^\top \right),$$

which represents the symmetric part of the input-output projection matrix $S_C^\top S_B$. It is worth noting that in case $D = 1$, $\mu = S_C^\top S_B$ as expected. The matrix $\mu$ has only real eigenvalues. We particularly conjecture that all tokens will converge to zero if $\mu$ has only negative eigenvalues, whereas the tokens will diverge to infinity if $\mu$ has at least one positive eigenvalue (see Appendix B for further detail). We leave the study of the higher-dimensional case for future work.

**Remark 4** (Compare with tokens' dynamic in Transformer). As outlined in (Geshkovski et al., 2024) (page 6, paragraph 2), tokens in a purely self-attention setting typically exhibit exponential divergence to infinity. However, our theoretical analysis demonstrates that the token dynamics in the purely Mamba setting are much more complicated. Specifically, tokens in the Mamba framework may either converge to zero or diverge to infinity. In cases of divergence, the behavior can vary: tokens may diverge rapidly to infinity within a finite time or diverge more gradually to infinity over an infinite time horizon, with the divergence following a logarithmic rate.

In the subsequent subsections, we provide the statements of the main theorems. The proofs of these theorems can be found in Appendix A.

### 4.2 CONVERGENCE SCENARIO: $\mu = S_C^\top S_B < 0$

In the following theorem, we prove that all tokens converge to zero at a rate of $O(1/\sqrt{t})$ as $t$ approaches infinity, regardless of the input data. Consequently, the hidden attention scores will also tend toward zero.

**Theorem 4.1** (Convergence scenario). *Assume that $\mu = S_C^\top S_B < 0$. Let $\mathbf{x}(t) = (x_1(t), \ldots, x_L(t))$ be the unique solution of the dynamic* (9).

1. *For each $l = 1, \ldots, L$, if $x_{l0} > 0$ (respectively, $x_{l0} < 0$), then $x_l(t)$ is positive and monotonically decreasing (respectively, negative and monotonically increasing) on $[0, +\infty)$. In addition, $x_l(t) = O(\frac{1}{\sqrt{t}})$.*

2. *$P_{lj}(t) = O(\frac{1}{t})$ for all $l, j = 1, \ldots, L$.*

The proof of this theorem can be found in Appendix A.1. To intuitively see why all tokens converge in this scenario, we can rewrite the dynamic (9) as

$$\frac{d}{dt}x_l(t) = \mu x_l(t)^3 \Delta(x_l(t)) + g_l(t)x_l(t), \tag{12}$$

where

$$g_l(t) = \begin{cases} 0, & \text{if } l = 1, \\ \displaystyle\sum_{j=1}^{l-1} \mu x_j(t)^2 \Delta(x_j(t)) \exp\left(-a \sum_{k=j+1}^{l} \Delta(x_k(t))\right), & \text{if } l > 1. \end{cases}$$

Since $\mu < 0$, $g_l(t)$ is a nonpositive function. Let us consider the case when $x_{l0} > 0$. Then, $x_l(t)$ must remain positive for every $t > 0$; otherwise, $x_l(t)$ would reach zero at some point, and consequently $x_l(t)$ would be identically zero for all $t$, according to the uniqueness of the given initial value problem, which is impossible. As a result, the right-hand side of equation (12) is always negative. This implies that $x_l(t)$ is decreasing. Figure 1 illustrates the graph of tokens and the heat map of the corresponding hidden attention matrices in this scenario.

**Remark 5** (Negative impact on model's performance). In this scenario, both the tokens and hidden attention scores collapse to zero, leaving no room for diversity or randomness, which may be harmful to the model's performance. Our experiment in Section 5.1 verifies these effects in high dimensions.

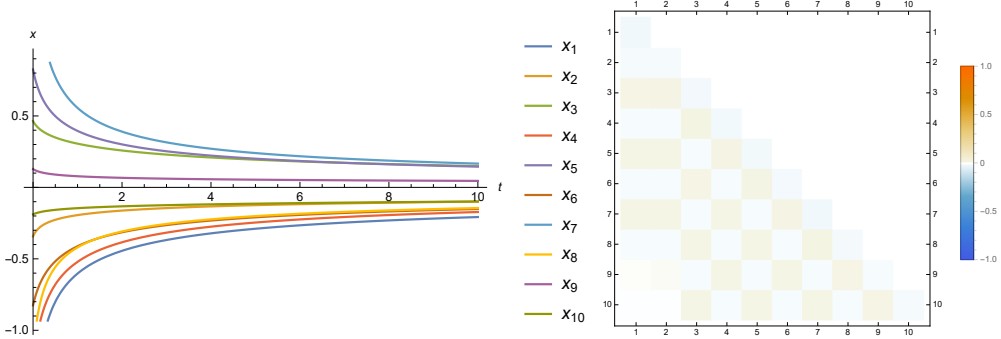

Figure 1: The graphs of tokens (left) and the heat map of the hidden attention matrices (right) for the convergence scenario with $L = 10$ tokens and parameters $\mu = -1.58$, $S_\Delta = -0.17$, $a = -1.08$, as well as initial data $\mathbf{x}(0) = (-1.79, -0.34, 0.46, -1.25, 0.83, -0.83, 1.81, -1.16, 0.13, -0.19)$. In this case, all tokens and hidden attention scores tend to zero as $t$ approaches infinity.

### 4.3 SLOW-DIVERGENCE SCENARIO: $\mu = S_C^\top S_B > 0$ AND $S_\Delta x_{l0} < 0$ FOR ALL $l$

In this case, we first observe that all of the tokens will tend to infinity as we will see in the following lemma.

**Lemma 4.2** (Slow divergence scenario). *Assume that $\mu = S_C^\top S_B > 0$ and $S_\Delta x_{l0} < 0$ for all $l$. Let $x(t) = (x_1(t), \ldots, x_L(t))$ be the unique solution of the dynamic (15). For each $l$, if $x_{l0} > 0$ (respectively, $x_{l0} < 0$), then $x_l(t)$ is positive and monotonically increasing (respectively, negative and monotonically decreasing) on $[0, +\infty)$ with*

$$\lim_{t \to +\infty} x_l(t) = +\infty \quad (\text{respectively, } \lim_{t \to +\infty} x_l(t) = -\infty).$$

Lemma 4.2 shows that all tokens in the considered case approach infinity as $t$ approaches infinity. However, it does not help to estimate the rate of divergence or the asymptotic behavior of the hidden attention score $P_{lj}(t)$. In Theorem 4.3 below, we provide the rate of divergence for the tokens and the asymptotic behavior of the hidden attention scores, but under a certain additional assumption.

**Theorem 4.3** (Slow divergence scenario and divergence rate). *Assume that $\mu = S_C^\top S_B > 0$ and*

$$S_\Delta x_{L0} \le \ldots \le S_\Delta x_{10} \le -2.12, \tag{13}$$

*(see Theorem A.7 for a detailed upper bound). Let $x(t) = (x_1(t), \ldots, x_L(t))$ be the unique solution of the dynamic (9).*

1. *For each l, if $x_{l0} > 0$ (respectively, $x_{l0} < 0$), then $x_l(t)$ is a positive increasing (respectively, negative decreasing) function on $[0, +\infty)$. In addition, $x_l(t) = O((\ln t)^l)$.*

2. $\lim_{t \to +\infty} P_{lj}(t) = 0$ *for all $l \geq j$.*

**Remark 6** (Unequally tokens' contribution). Theorem 4.3 shows that, in the one-dimensional case within this divergence scenario, tokens with higher absolute values will diverge to infinity faster than those with smaller absolute values (while all hidden attention scores still converge to zero). Intuitively speaking, this means that different tokens will contribute differently to the updates during training. Based on this observation, we conjecture that reordering tokens based on their importance scores before training might be beneficial for model performance. Our experiment in Subsection 5.2 empirically confirms this conjecture.

The proof of this theorem can be found in Appendix A.2. We can intuitively see why the tokens tend to infinity slowly by looking at the values of the right-hand side of equation (12). Indeed, let us assume that $S_\Delta < 0$ and thus $x_{l0} > 0$. Then $x_l(t)$ must always be positive as $t$ increases; otherwise, $x_l(t)$ would reach zero at some point, and consequently $x_l(t)$ would be identically zero for all $t$, according to the uniqueness of the given initial value problem, which is impossible. As a result, the right-hand side of equation (12) is always not too large. This implies that $x_l(t)$ is decreasing. However, since $S_\Delta < 0$, the term $\mu x_l(t)^3 \Delta(x_l(t))$ converges to zero very quickly, which keeps the right-hand side always not too large. Therefore, $x_l(t)$ goes to infinity slowly. Figure 2 illustrates the graph of tokens and the heat map of the corresponding hidden attention matrices in this scenario.

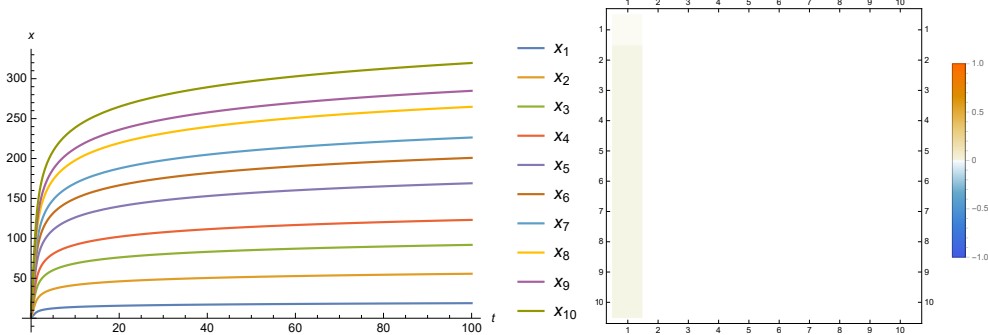

Figure 2: The graphs of tokens (left) and the heat map of the hidden attention matrices (right) for the convergence scenario with $L = 10$ tokens and the model parameters $\mu = 1.79$, $S_\Delta = -0.71$, $a = -1.80$, as well as the initial data $\mathbf{x}(0) = (1.55, 2.84, 3.81, 4.57, 5.99, 6.94, 7.71, 8.96, 9.59, 10.75)$. In this case, tokens tend to infinity at the log-rate, and the one which larger initial value diverges faster, while the hidden attention scores tend to zero as $t$ approaches infinity. In addition, the hidden attention scores from the second columns tend to zero must faster than those in the first column.

### 4.4 FAST-DIVERGENCE SCENARIO: $\mu = S_C^\top S_B > 0$ AND $S_\Delta x_{l0} > 0$ FOR SOME $l$

In the fast divergence scenario, we have the following theorem.

**Theorem 4.4** (Fast divergence scenario). *Assume that $\mu = S_C^\top S_B > 0$. Let $x(t) = (x_1(t), \ldots, x_L(t))$ be the unique solution of the dynamic (15). If there exists $l = 1, \ldots, L$ such that $S_\Delta x_{l0} > 0$, then $x_l(t)$ goes to infinity at finite time.*

*In addition, assume that $x_{l_0}(t)$ tends to infinity first as $t \to T^-$ for some $T > 0$ while $x_l(t)$ is defined on $[0, T]$ for all $l \neq l_0$. Then for every $L \geq l \geq j \geq 1$, $\lim_{t \to T^-} P_{lj}(t) = +\infty$ if and only if $l = l_0$ or $j = l_0$.*

**Remark 7.** The proof of this theorem can be found in Appendix A.3. Intuitively speaking, in this scenario, if $1 \leq l \leq L$ is the index of the first token such that $S_\Delta x_{l0} > 0$, then all tokens with indices $j \geq l$ will diverge to infinity in finite time. Figure 3 illustrates the dynamical properties of tokens and the hidden attention scores in this scenario.

## 5 EXPERIMENTS

In this section, we empirically validate two key observations derived from the theoretical analysis presented in Section 4. We aim to (i) demonstrate the detrimental impact of negative eigenvalues in

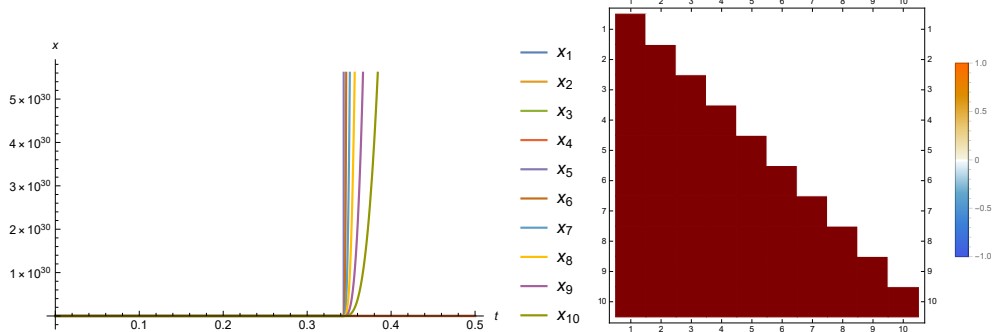

Figure 3: The graphs of tokens (left) and the heat map of the hidden attention matrices (right) for the convergence scenario with $L = 10$ tokens and the model parameters $\mu = 0.76$, $S_\Delta = 0.59$, $a = 1.66$, as well as the initial data $\mathbf{x}(0) = (0.83, 0.91, 0.64, 0.78, 0.66, 0.99, 0.68, 0.72, 0.61, 0.90)$. In this case, tokens and the hidden attention scores tend to infinity very quickly at finite time.

the input-output matrix $S_C^\top S_B$ on the performance of auto-regressive language modeling tasks and (ii) propose a reordering strategy and illustrate its effectiveness in vision tasks. Details on datasets, models, and training procedures are provided in Appendix C.

### 5.1 EFFECTS OF NEGATIVE EIGENVALUES OF INPUT-OUTPUT MATRIX

We use Mamba (Gu & Dao, 2024) as the baseline and conduct evaluations on the WIKITEXT103 language modeling task (Merity et al., 2017). We consider three distinct scenarios, including cases where the input-output matrix contains (i) only positive eigenvalues, (ii) only negative eigenvalues, as discussed in Section 4 and Appendix B, and (iii) both positive and negative eigenvalues as an intermediate (mixed) case. The scenario where $S_C^\top S_B$ encompasses complex eigenvalues is deferred to future work. We report the perplexity (PPL) for each scenario. Lower PPL values indicate better model performance.

Table 2: Test perplexity on WIKI-TEXT103. A model where $S_C^\top S_B$ has a smaller proportion of negative eigenvalues achieves lower perplexity.

| Scenario | Perplexity ($\downarrow$) |
|---|---|
| Negative | 17.28 |
| Mixed | 16.82 |
| Positive | 16.66 |

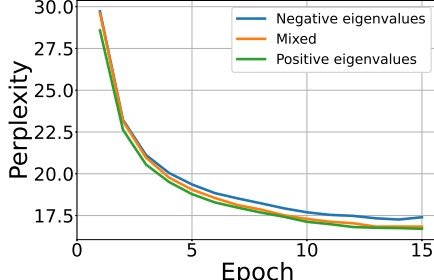

Figure 4: Test perplexity on WIKITEXT103 during training procedure. The positive case consistently demonstrates superior performance compared to the other two scenarios.

Table 2 and Figure 4 illustrate the impact of different eigenvalue configurations on the model's performance. Specifically, our findings demonstrate a correlation between the prevalence of positive eigenvalues in the input-output matrix and model performance. Conversely, the presence of negative eigenvalues is linked to a decrease in performance, suggesting that removing negative eigenvalues could further optimize the model's efficiency.

### 5.2 RE-ORDERING INPUT TOKENS

We now emphasize the significance of token ordering in vision tasks, where tokens naturally lack an inherent structure due to each token representing a patch from a 2D or 3D image. As indicated in Theorem 4.3, the contribution of each token to model updates is primarily influenced by the term $S_\Delta x_{l0}$, which is a vector of dimension $D$ in a scenario with $D$ channels. To determine the optimal order of tokens $x_{l0}$, we define an importance score as follows:

$$s_l = \langle K, S_\Delta x_{l0} \rangle \tag{14}$$

where $K \in \mathbb{R}^D$ is a learnable parameter. Prior to passing the tokens through the S6 layer, they are reordered based on their importance scores in ascending order. We use SoftSort (Prillo & Eisenschlos, 2020) to enable gradient propagation through the sorting operation. Specifically, SoftSort is employed to generate a permutation matrix, which is then applied to sort the $s_l$ array and rearrange the tokens $x_{l0}$ accordingly. Details of the procedure and an analysis of the computational cost are provided in Appendix D.2.

To assess the effectiveness of our proposed reordering technique, we benchmark on the image classification task using the ImageNet-1K dataset (Deng et al., 2009). We employ MambaVision (Hatamizadeh & Kautz, 2024) as the baseline and compare the top-1 and top-5 accuracy metrics.

Table 3: Top-1 and Top-5 accuracy ($\uparrow$) on the large-scale ImageNet-1K image classification task. Our token reordering method leads to a slight improvement in MambaVision's performance.

| Model | Top-1 (%) | Top-5 (%) |
| --- | --- | --- |
| MambaVision-T (baseline) | 81.90 | 95.86 |
| MambaVision-T + Token reordering | **82.02** | **95.87** |

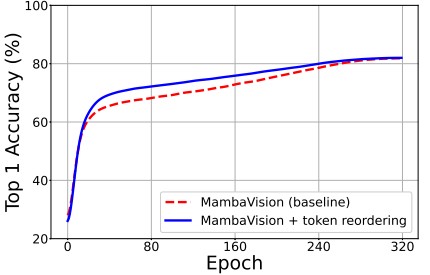 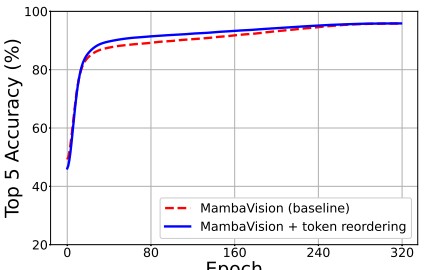

Figure 5: Top-1 (left) and Top-5 (right) accuracy ($\uparrow$) on ImageNet-1K during the training process. Our token reordering method boosts the accuracy of the MambaVision baseline and achieves faster convergence compared to the baseline.

The results presented in Figure 5 and Table 3 demonstrate that our reordering method speeds up the convergence of the training loss and improves the performance of the MambaVision baseline. These findings confirm the effectiveness of our technique in vision tasks, highlighting its potential for broader tasks. Experiments on language modeling and text classification are in Appendix D.1.

## 6 CONCLUDING REMARKS

We characterize the dynamical properties of tokens in a pre-trained Mamba model. In the one-dimensional case, we show that either all tokens converge to zero or all tokens diverge to infinity. In addition, for the convergent scenario, we empirically verify that this scenario negatively impacts the model's performance; therefore, it should be excluded from model training. For the divergent scenario, we prove that tokens at different locations will diverge to infinity at different rates, thereby contributing unequally to the updates during model training. Based on these investigations, we propose two refinements for the model: excluding the convergent scenario and reordering tokens based on their importance scores, both aimed at improving practical performance. Our experimental results suggest that these refinements do help improve model performance.

Regarding limitations, our theoretical analysis of the dynamical properties of tokens is restricted to tokens with one feature channel. In addition, our token reordering refinement, based on importance scores, introduces additional computational overhead during training, although this increase is relatively small. Furthermore, we are uncertain whether the method of reordering tokens based on their orthogonal projection onto a learnable affine line, as used in practical implementation, is optimal. We leave a systematic study of token reordering refinements as well as generalization of our theoretical analysis of higher-dimensional cases for future work.

ACKNOWLEDGMENTS

This research / project is supported by the National Research Foundation Singapore under the AI Singapore Programme (AISG Award No: AISG2-TC-2023-012-SGIL). This research / project is supported by the Ministry of Education, Singapore, under the Academic Research Fund Tier 1 (FY2023) (A-8002040-00-00, A-8002039-00-00). This research / project is supported by the NUS Presidential Young Professorship Award (A-0009807-01-00) and the NUS Artificial Intelligence Institute–Seed Funding (A-8003062-00-00). This research / project is also supported by Singapore National Academy of Science under the SASEA Fellowship Programme (Award No: NRF-MP-2025-0001).

**Reproducibility Statement.** Our implementation details are provided in Section 5 and Appendix C. Source code is provided at https://github.com/Fsoft-AIC/Mamba-token-dynamic.

**Ethics Statement.** Considering the scope of our research, we do not anticipate any negative societal or ethical consequences emerging from this work.

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

# Supplement to "Demystifying the Token Dynamics of Deep Selective State Space Models"

## A  Dynamical Properties Tokens in Mamba

We will consider S6 dynamic in the single channel dimension case, i.e. $D = 1$, in our setting to avoid overly complicated technicalities. In this case, the input sequence $\mathbf{x} = (x_1, \ldots, x_L) \in \mathbb{R}^L$ is a list of $L$ real numbers. The dynamic of $\mathbf{x}$ is then governed by the dynamical system:

$$\frac{d}{dt}x_l(t) = \sum_{j=1}^{l} P_{lj}(t)x_j(t), \quad t \in [0, +\infty), \tag{15}$$

$$x_l(0) = x_{l0} \in \mathbb{R},$$

for $l = 1, \ldots, L$, where

$$\Delta(u) = \text{softplus}(S_\Delta u) = \ln\left(1 + e^{S_\Delta u}\right), \qquad u \in \mathbb{R}, \tag{16}$$

and

$$P_{lj}(t) = \begin{cases} \left(S_C^\top S_B\right) x_l(t)^2 \cdot \Delta(x_l(t)), & \text{if } l = j, \\ \left(S_C^\top S_B\right) x_l(t)x_j(t) \cdot \Delta(x_j(t)) \cdot \exp\left(-a \sum_{k=j+1}^{l} \Delta(x_k(t))\right), & \text{if } l > j. \end{cases} \tag{17}$$

In this case, $S_B, S_C \in \mathbb{R}^{N \times 1}$, thus $S_C^\top S_B \in \mathbb{R}$, and the scalar values $S_\Delta$ and $a \in \mathbb{R}$, with $a > 0$, are learnable from input data.

To avoid triviality, we will always assume that the initial datum $x_l(0) = x_{l0}$ are nonzero for all $l$. The main goal of this section is to investigate the asymptotic behavior of the solution $\mathbf{x}(t) = (x_1(t), \ldots, x_L(t))$ of the dynamic (15) and the corresponding hidden attention matrix

$$\mathbf{P}(t) = \begin{bmatrix} P_{11}(t) & 0 & \ldots & 0 \\ P_{21}(t) & P_{22}(t) & \ldots & 0 \\ \ldots & \ldots & \ldots & \ldots \\ P_{L1}(t) & P_{L2}(t) & \ldots & P_{LL}(t) \end{bmatrix},$$

when $t$ increases. The following three scenarios will be considered separately in the subsequent subsections:

1. Convergence scenario: $\mu = S_C^\top S_B < 0$;

2. Slow-divergence scenario: $\mu = S_C^\top S_B > 0$ and $S_\Delta x_{l0} < 0$ for all $l$; and

3. Fast-divergence scenario: $\mu = S_C^\top S_B > 0$ and $S_\Delta x_{l0} > 0$ for some $l$.

### A.1  Convergence Scenario: $\mu = S_C^\top S_B < 0$

We will start with the following auxiliary lemmas.

**Lemma A.1.** *Let $t_0 \in \mathbb{R}$ and let $f(u, t)$ be a differentiable continuous function on $\mathbb{R} \times [t_0, +\infty)$. Assume that $u(t)$ is the unique solution of the initial value problem*

$$\frac{d}{dt}u(t) = f(u(t), t)\, u(t), \quad t \in [t_0, +\infty),$$

$$u(t_0) = u_0 \in \mathbb{R}.$$

*If $u_0 > 0$ (respectively, $u_0 < 0$), then $u(t)$ is a positive (respectively, negative) function on its maximal interval of the existence. In addition,*

1. *if $f$ is a positive function, then $u(t)$ is monotonically increasing (respectively, monotonically decreasing),*

    *2. if $f$ is a negative function, then $u(t)$ is monotonically decreasing (respectively, monotonically increasing).*

*Proof.* We assume that $u_0 > 0$. The case when $u_0 < 0$ is obtained by replacing $u(t)$ by $-u(t)$. The function $u(t)$ cannot be equal to zero on its maximal interval, since otherwise, $u(t) = 0$ is the unique solution of the given differential equaion, which contradicts to the fact that $u(0) = u_0 > 0$. Therefore, $u(t)$ is positive on its maximal interval of the existence.

Next, assume that $f$ is positive on $\mathbb{R} \times [t_0, +\infty)$. Then we always have $u'(t) = f(u(t), t)u(t) > 0$ for all $t \in [t_0, +\infty)$. This shows that $u(t)$ is increasing on its maximal interval of the existence. The case when $f$ is negative is similar. $\qquad\square$

**Lemma A.2.** *Let $t_0 \in \mathbb{R}$. If a function $u(t)$ is bounded on $[t_0, +\infty)$, then there exists positive numbers $c_1, c_2$ such that*

$$c_1 \leq \Delta(u(t)) \leq c_2,$$

*for all $t \in [t_0, +\infty)$.*

*Proof.* Recall that $\Delta(u(t)) = \ln(1 + e^{S_\Delta u(t)})$. Since $u(t)$ is bounded, there are $a_1, a_2 \in \mathbb{R}$ such that $a_1 \leq S_\Delta u(t) \leq a_2$ for all $t \in [t_0, +\infty)$. Then we can choose $c_i = \ln(1 + e^{a_i})$ for $i = 1, 2$. $\qquad\square$

**Theorem A.3** (Convergence scenario). *Assume that $\mu = S_C^\top S_B < 0$. Let $\mathbf{x}(t) = (x_1(t), \ldots, x_L(t))$ be the unique solution of the dynamic* (15)*.*

    *1. For each $l = 1, \ldots, L$, if $x_{l0} > 0$ (respectively, $x_{l0} < 0$), then $x_l(t)$ is positive and monotonically decreasing (respectively, negative and monotonically increasing) on $[0, +\infty)$. In addition, $x_l(t) = O(\frac{1}{\sqrt{t}})$.*

    *2. $P_{lj}(t) = O(\frac{1}{t})$ for all $l, j = 1, \ldots, L$.*

*Proof.* Assume without loss of generality that $x_{l0} > 0$. The case when $x_{l0} < 0$ is proved similarly by replacing $x_l(t)$ by $-x_l(t)$. We rewrite the dynamic (15) as

$$\frac{d}{dt}x_l(t) = \mu x_l(t)^3 \Delta(x_l(t)) + g_l(t)x_l(t), \qquad (18)$$

where

$$g_l(t) = \begin{cases} 0, & \text{if } l = 1, \\ \displaystyle\sum_{j=1}^{l-1} \mu x_j(t)^2 \Delta(x_j(t)) \exp\left(-a \sum_{k=j+1}^{l} \Delta(x_k(t))\right), & \text{if } l > 1. \end{cases}$$

Since $\mu < 0$, $g_l(t)$ is a nonpositive function. According to Lemma A.1, $x_l(t)$ is positive and monotonically decreasing on its right maximal interval of the existence $[0, T)$ with $T \in (0, +\infty]$. Thus, $x_l(t)$ is bounded and $T = +\infty$. As a consequence, it follows from Lemma A.2 that there is a positive constant $c > 0$ such that $\Delta(x_l(t)) \geq c$ for all $t \in [0, +\infty)$. Therefore, from Eq. (18), we have

$$\frac{d}{dt}x_l(t) \leq \mu c x_l(t)^3,$$

which imply that $x_l(t) \leq (-2\mu ct + x_{l0}^{-2})^{-\frac{1}{2}}$ for all $t \in [0, +\infty)$. Hence, $x_l(t) = O(\frac{1}{\sqrt{t}})$ as expected.

For the hidden attention score $P_{lj}(t)$ with $L \geq l \geq j \geq 1$, we have

$$|P_{lj}(t)| = \mu |x_l(t)x_j(t)| \Delta(x_j(t)) \exp\left(-a \sum_{k=j+1}^{l} \Delta(x_k(t))\right) \leq \mu c |x_l(t)x_j(t)| = O\left(\frac{1}{t}\right),$$

as expected. $\qquad\square$

## A.2    SLOW-DIVERGENCE SCENARIO: $\mu = S_C^\top S_B > 0$ AND $S_\Delta x_{l0} < 0$ FOR ALL $l$

In this case, we first observe that all of the tokens will tend to infinity as we will see in the following lemma.

**Lemma A.4** (Slow-divergence scenario). *Assume that $\mu = S_C^\top S_B > 0$ and $S_\Delta x_{l0} < 0$ for all $l$. Let $x(t) = (x_1(t), \ldots, x_L(t))$ be the unique solution of the dynamic (15). For each $l$, if $x_{l0} > 0$ (respectively, $x_{l0} < 0$), then $x_l(t)$ is positive and monotonically increasing (respectively, negative and monotonically decreasing) on $[0, +\infty)$ with*

$$\lim_{t \to +\infty} x_l(t) = +\infty \quad (\text{respectively, } \lim_{t \to +\infty} x_l(t) = -\infty).$$

*Proof.* Without loss of generality, we assume that $S_\Delta < 0$. The case when $S_\Delta > 0$ is then obtained by replacing $x(t)$ by $-x(t)$. Similar to the proof of Theorem A.3, we rewrite dynamic (15) as

$$\frac{d}{dt} x_l(t) = \mu x_l(t)^3 \Delta(x_l(t)) + g_l(t) x_l(t), \tag{19}$$

where

$$g_l(t) = \begin{cases} 0, & \text{if } l = 1, \\ \sum_{j=1}^{l-1} \mu x_j(t)^2 \Delta(x_j(t)) \exp\left(-a \sum_{k=j+1}^{l} \Delta(x_k(t))\right), & \text{if } l > 1. \end{cases} \tag{20}$$

Since $\mu > 0$, $g_l(t)$ is a nonnegative function on $[0, +\infty)$.

According to Lemma A.1, $x_l(t)$ must be a positive increasing function on its maximal interval of the existence $[0, T)$ for some $T \in (0, +\infty]$.

First, we claim that $x_l(t)$ is unbounded on $[0, T)$. Indeed, if this is not the case, then $x_l(t)$ is bounded and $T = +\infty$. According to Lemma A.2, there exists a positive constant $c$ such that $\Delta(x_l(t)) \geq c$ for all $t \in [0, +\infty)$. Therefore, Eq. (19) yields

$$\frac{d}{dt} x_l(t) \geq \mu c x_l(t)^3, \quad t \in [0, +\infty),$$

which implies that $x_l(t) \geq (-2\mu c t + x_{l0}^{-2})^{-\frac{1}{2}}$, which is unbounded at $t = \frac{1}{c\mu x_{l0}^2} > 0$, a contradiction. The claim is then proved, and $\lim_{t \to T^-} x_l(t) = +\infty$.

It remains to prove that $T = +\infty$. Observe that, since $S_\Delta < 0$, we have

$$\lim_{r \to +\infty} r^3 \Delta(r) = \lim_{r \to +\infty} \frac{r^3}{e^{-S_\Delta r}} \frac{\ln(1 + e^{S_\Delta r})}{e^{S_\Delta r}} = 0.$$

Furthermore, since $x_l(t)$ is positive and monotonically increasing to infinity at infinity, we also have

$$\lim_{t \to +\infty} x_l(t)^3 \Delta(x_l(t)) = 0.$$

In particular, in case $l = 1$, we have $\frac{d}{dt} x_1(t) \leq 1$, which implies that $x_1(t) \leq t$, for all $t$ large enough. Therefore, $T = +\infty$ in this case.

Assume that $l > 1$ and we already have $T = +\infty$ for all cases $j < l$. Then it follows from Eq (20) that

$$g_l(t) \leq \sum_{j=1}^{l-1} \mu x_j(t)^2 \Delta(x_j(t)) \leq \sum_{j=1}^{l-1} \frac{\frac{d}{dt} x_j(t)}{x_j(t)} = \frac{d}{dt} \sum_{j=1}^{l-1} \ln(x_j(t)). \tag{21}$$

Therefore, we can bound $\frac{d}{dt} x_l(t)$ from Eq. (19) as

$$\frac{d}{dt} x_l(t) \leq 1 + \left(\frac{d}{dt} \sum_{j=1}^{l-1} \ln(x_j(t))\right) x_l(t),$$

and this inequality hold for all $t \geq t_1$ for some $t_1$ large enough. As a consequence, we have

$$x_l(t) \leq \exp\left(\sum_{j=1}^{l-1} \ln(x_j(t)) - \ln(x_j(t_1))\right) \left(\int_{t_1}^{t} \exp\left(-\sum_{j=1}^{l-1} \ln(x_j(s)) - \ln(x_j(t_1))\right) ds + x_l(t_1)\right).$$

The right hand side is a function defined for all $t \geq t_1$. Thus $T = +\infty$. $\square$

**Remark 8.** Lemma A.4 shows that all tokens in the considered case will approach infinity as $t$ approaches infinity. However, it does not help to estimate the rate of divergence or the asymptotic behavior of the hidden attention score $P_{lj}(t)$. In Theorem A.7 below, we will provide both the rate of divergence for the tokens and the asymptotic behavior of the hidden attention scores, but under a certain additional assumption.

We provide the divergence rate of the first token in the following lemma.

**Lemma A.5.** *Assume that $\mu = S_C^\top S_B > 0$ and $S_\Delta x_{10} < 0$. Let $x_1(t)$ be the unique solution of the dynamic* (15). *Then $x_1(t)$ is defined on $[0, +\infty)$ and $x_1(t) = O(\ln(t))$.*

*Proof.* Without loss of generality, let us assume that $S_\Delta < 0$, thus $x_{10} > 0$. The case when $S_\Delta > 0$ can be obtained by replacing $x_1(t)$ by $-x_1(t)$. Then it follows from Lemma A.4 that $x_1(t)$ is a positive and monotonically increasing function on $[0, +\infty)$. Let us fix a small enough $\epsilon > 0$ such that $S_\Delta + \epsilon < 0$. Then, there exist $c_1, t_1 > 0$ such that:

$$\mu x_1(t)^3 \Delta(x_1(t)) = e^{(S_\Delta + \epsilon)x_1(t)} \cdot \frac{\mu x_1(t)^3}{e^{\epsilon x_1(t)}} \cdot \frac{\ln(1 + e^{S_\Delta x_1(t)})}{e^{S_\Delta x_1(t)}} \le c_1 e^{(S_\Delta + \epsilon)x_1(t)}, \qquad (22)$$

for all $t \in [t_1, +\infty)$. Therefore, we have

$$\frac{d}{dt}x_1(t) \le c_1 e^{(S_\Delta + \epsilon)x_1(t)}, \quad t \ge t_1,$$

which yields

$$x_1(t) \le -\frac{2}{S_\Delta} \ln\left(-\frac{1}{2}S_\Delta c_1(t - t_1) + e^{-\frac{1}{2}S_\Delta x_1(t_1)}\right), \quad \text{for all } t \ge t_1.$$

Since $x_1(t)$ is positive, we finally obtain $x_1(t) = O(\ln(t))$ as expected. $\qquad \square$

To determine the divergence rate of the other tokens, we will need the following lemma.

**Lemma A.6.** *Let $r_0(\approx 2.1160)$ be the unique positive root of the function $h(r) = 2\ln(1 + e^{-r}) - \frac{re^{-r}}{1+e^{-r}}$. Then for $\lambda > 0$, the function $f(r) = r^2 \ln\left(1 + e^{-\lambda r}\right)$ is positive and monotonically decreasing on $[\frac{r_0}{\lambda}, +\infty)$.*

*Proof.* The lemma follows from the negativity of the function $h(r)$ on $[r_0, +\infty)$ and the equality $\frac{d}{dt}f(r) = rh(\lambda r)$. $\qquad \square$

**Theorem A.7** (Slow divergence scenario and divergent rate). *Assume that $\mu = S_C^\top S_B > 0$ and*

$$S_\Delta x_{L0} \le \ldots \le S_\Delta x_{10} \le -r_0, \qquad (23)$$

*where $r_0(\approx 2.1160)$ is defined in Lemma A.6. Let $x(t) = (x_1(t), \ldots, x_L(t))$ be the unique solution of the dynamic* (15).

  1. *For each $l$, if $x_{l0} > 0$ (respectively, $x_{l0} < 0$), then $x_l(t)$ is a positive increasing (respectively, negative decreasing) function on $[0, +\infty)$. In addition, $x_l(t) = O((\ln t)^l)$.*

  2. $\lim_{t \to +\infty} P_{lj}(t) = 0$ *for all $l \ge j$.*

*Proof.* Without loss of generality, we will assume that $S_\Delta = -\lambda$ for some $\lambda > 0$. The case when $S_\Delta > 0$ is then obtained by changing all $x_l$ to $-x_l$. Then the condition (23) becomes

$$x_{L0} \ge \ldots \ge x_{10} \ge \frac{r_0}{\lambda}. \qquad (24)$$

According to Proposition A.4, $x_l(t)$ is a positive and monotonically increasing function on $[0, +\infty)$ with $\lim_{t \to +\infty} x_l(t) = +\infty$.

Before estimating the asymptotic behavior of $x_l(t)$ and $P_{lj}(t)$, we first claim that that $x_l(t) \ge x_{l-1}(t)$ for all $t \in [0, +\infty)$ and $l \ge 2$. We will prove this claim by induction on $l$. Indeed, in case $l = 2$, we observe that

$$\frac{d}{dt}x_2(t) = \mu x_2(t)^3 \Delta(x_2(t)) + g_2(t)x_2(t) \ge \mu x_2(t)^3 \Delta(x_2(t)).$$

Since $x_{20} \geq x_{10}$, it follows that $x_2(t) \geq x_1(t)$ for all $t \in [0, +\infty)$. Let us assume that $l > 2$ and $x_{l-1}(t) \geq \ldots \geq x_1(t)$ for all $t \in [0, +\infty)$. It is noted that $x_1(t) \geq x_{10} \geq \frac{r}{\lambda}$. According to Lemma A.6, we have

$$x_j(t)^2 \Delta(x_j(t)) \geq x_{j-1}(t)^2 \Delta(x_{j-1}(t)), \quad j = 2, \ldots, l-1. \tag{25}$$

Recall that

$$\frac{d}{dt} x_l(t) = \mu x_l(t)^3 \Delta(x_l(t)) + g_l(t) x_l(t) \tag{26}$$

$$= \mu x_l(t)^3 \Delta(x_l(t)) + \mu x_l(t) \sum_{j=1}^{l-1} x_j(t)^2 \Delta(x_j(t)) \exp\left(-a \sum_{k=j+1}^{l-1} \Delta(x_k(t))\right). \tag{27}$$

By removing the term with the index $j = 1$ inside the sum and using inequality (25), we obtain

$$\frac{d}{dt} x_l(t) \geq \mu x_l(t)^3 \Delta(x_l(t)) + \mu x_l(t) \sum_{j=2}^{l-1} x_{j-1}(t)^2 \Delta(x_{j-1}(t)) \exp\left(-a \sum_{k=j+1}^{l-1} \Delta(x_k(t))\right)$$

$$= \mu x_l(t)^3 \Delta(x_l(t)) + \mu x_l(t) \sum_{j=1}^{l-2} x_j(t)^2 \Delta(x_j(t)) \exp\left(-a \sum_{k=j+2}^{l-1} \Delta(x_k(t))\right).$$

Since $\Delta(x_k(t)) \leq \Delta(x_{k-1}(t))$ for all $k = 2, \ldots, l-1$, we can proceed the last expression as

$$\frac{d}{dt} x_l(t) \geq \mu x_l(t)^3 \Delta(x_l(t)) + \mu x_l(t) \sum_{j=1}^{l-2} x_j(t)^2 \Delta(x_j(t)) \exp\left(-a \sum_{k=j+1}^{l-2} \Delta(x_k(t)) - a\Delta(x_l(t))\right).$$

Therefore, if we set

$$f(u) = \mu u^3 \Delta(u) + \mu u \sum_{j=1}^{l-2} x_j(t)^2 \Delta(x_j(t)) \exp\left(-a \sum_{k=j+1}^{l-2} \Delta(x_k(t)) - a\Delta(u)\right),$$

then $x_{l-1}(t)$ is a solution of the differential equation $\frac{d}{dt} u(t) = f(u(t))$, while $\frac{d}{dt} x_l(t) \geq f(x_l(t))$ for all $t \in [0, +\infty)$. It follows from the initial conditions $x_{l0} \geq x_{l-1,0}$ that $x_l(t) \geq x_{l-1}(t)$ for all $t \in [0, +\infty)$. The claim is then proved.

Next, let us go back to the analysis of the asymptotic behaviour of tokens and hidden attention scores.

1. For $l = 1$, it is already known from Lemma A.5 that $x_1(t) = O(\ln(t))$. In case $l \geq 2$, according to equation (27), we have

$$\frac{\frac{d}{dt} x_l(t)}{x_l(t)} = \mu x_l(t)^2 \Delta(x_l(t)) + \sum_{j=1}^{l-1} \mu x_j(t)^2 \Delta(x_j(t)) \exp\left(-a \sum_{k=j+1}^{l-1} \Delta(x_k(t))\right) \tag{28}$$

For each $j = 1, \ldots, l$, since $x_j(t) \geq x_1(t)$ and the function $r \mapsto r^2 \Delta(r)$ is monotonically decreasing over $[\frac{r_0}{\lambda}, +\infty)$, we have $x_j(t)^2 \Delta(x_j(t)) \leq x_1(t)^2 \Delta(x_1(t))$ for all $t > 0$ large enough. In addition, we also have and $\exp\left(-a \sum_{k=j+1}^{l-1} \Delta(x_k(t))\right) \leq 1$ for all $t \in [0, +\infty)$. Therefore, we can upper bound the right hand side of equation (28) as

$$\frac{\frac{d}{dt} x_l(t)}{x_l(t)} \leq l\mu x_1(t)^2 \Delta(x_1(t)) = l \frac{\frac{d}{dt} x_1(t)}{x_1(t)}. \tag{29}$$

Thus, $\ln x_l(t) \leq l \ln x_1(t)$, which implies that

$$x_l(t) \leq x_1(t)^l = O((\ln t)^l).$$

2. For the hidden attention score, it follows from its definition in equation (17) that

$$P_{lj}(t) \leq \mu x_l(t) x_j(t) \Delta(x_j(t))$$

for all $t \geq 0$. Since $S_\Delta < 0$, the function $r \mapsto r\Delta(r) = r\ln(1 + e^{S_\Delta r})$ is monotonically decreasing from a large enough $t$. Therefore, it follows from the claim above that

$$P_{lj}(t) \leq \mu x_l(t)^2 \Delta(x_l(t))$$

for all $t$ large enough. Hence, $\lim_{t \to +\infty} P_{lj}(t) = \lim_{r \to +\infty} \mu r^2 \Delta(r) = 0$.

The theorem is then proved. □

## A.3 FAST-DIVERGENCE SCENARIO: $\mu = S_C^\top S_B > 0$ AND $S_\Delta x_{l0} > 0$ FOR SOME $l$

**Theorem A.8** (Fast-divergence scenario). *Assume that* $\mu = S_C^\top S_B > 0$. *Let* $x(t) = (x_1(t), \ldots, x_L(t))$ *be the unique solution of the dynamic* (15). *If there exists* $l = 1, \ldots, L$ *such that* $S_\Delta x_{l0} > 0$, *then* $x_l(t)$ *goes to infinity at finite time.*

*In addition, assume that* $x_{l_0}(t)$ *tends to infinity first as* $t \to T^-$ *for some* $T > 0$ *while* $x_l(t)$ *is defined on* $[0, T]$ *for all* $l \neq l_0$. *Then for every* $L \geq l \geq j \geq 1$, $\lim_{t \to T^-} P_{lj}(t) = +\infty$ *if and only if* $l = l_0$ *or* $j = l_0$.

*Proof.* Without loss of generality, we assume that $S_\Delta > 0$. The case when $S_\Delta < 0$ is then obtained by replacing $x(t)$ by $-x(t)$. Then $x_{l0} > 0$. Recall that

$$\frac{d}{dt} x_l(t) = \mu x_l(t)^3 \Delta(x_l(t)) + g_l(t) x_l(t), \tag{30}$$

where $g_l(t)$ defined in equation (20). Since $\mu > 0$, $g_l(t)$ is a nonnegative function on $[0, +\infty)$. It follows from Lemma A.1 that $x_l(t)$ is a positive increasing function on its right maximal interval of the existence $[0, T)$ for some $T \in (0, +\infty]$. By equation (30), we have

$$\frac{d}{dt} x_l(t) \geq \mu x_l(t)^3 \Delta(x_l(t)).$$

Furthermore, since $S_\Delta > 0$, the function $\Delta(r) = \ln(1 + e^{S_\Delta r})$ is monotonically increasing on $[0, +\infty)$. Therefore, $\Delta(x_l(t)) \geq \Delta(x_l(0)) = \ln(1 + e^{S_\Delta x_{l0}})$ for all $t \in [0, +\infty)$. As a consequence, we can proceed the above inequality further as

$$\frac{d}{dt} x_l(t) \geq \mu \ln(1 + e^{S_\Delta x_{l0}}) x_l(t)^3.$$

Thus,

$$x_l(t) \geq \left(2\mu \ln(1 + e^{S_\Delta x_{l0}})t - x_{l0}^{-2}\right)^{-\frac{1}{2}},$$

which goes to $+\infty$ at finite time.

Next, let us assume that $x_{l0}(t)$ goes to infinity at finite time faster than the other tokens, i.e, there exists $T > 0$ such that $\lim_{t \to T^-} x_{l_0}(t) = +\infty$, while $x_l(t)$ defined over $[0, T]$ for all $l \neq l_0$. Then $\lim_{t \to T^-} x_{l_0}^2(t)\Delta(x_l(t)) = +\infty$. Then for every $L \geq l \geq j \geq 1$, $\lim_{t \to T^-} P_{lj}(t) = \infty$ if and only if $l = l_0$ or $j = l_0$. □

## B HIGHER DIMENSION WITH/WITHOUT ADDITIONAL COMPONENTS

When the dimension of the input tokens is $D \geq 1$, the input-output projection matrix $S_C^\top S_B$ is a square matrix of size $D \times D$, which may have complex eigenvalues. We conjecture that the dynamic behavior of the tokens in system (7) can be determined by analyzing the matrix

$$\mu = \frac{1}{2}\left(S_C^\top S_B + (S_C^\top S_B)^\top\right),$$

which represents the symmetric part of the input-output projection matrix $S_C^\top S_B$. This conjecture is motivated by the observation that the term $x_l^\top \cdot (S_C^\top S_B) \cdot x_l$ in the considered dynamical system (7) is a scalar, and it can be expressed as:

$$x_l^\top \cdot (S_C^\top S_B) \cdot x_l = \frac{1}{2} \left( x_l^\top \cdot (S_C^\top S_B) \cdot x_l + \left( x_l^\top \cdot (S_C^\top S_B) \cdot x_l \right)^\top \right) = x_l^\top \cdot \mu \cdot x_l.$$

The matrix $\mu$ has only real eigenvalues. We particularly conjecture that all tokens will converge to zero if $\mu$ has only negative eigenvalues, whereas the tokens will diverge to infinity if $\mu$ has at least one positive eigenvalue.

Figures 6 illustrate this conjecture by visualizing the trajectories of four tokens in the two-dimensional case under different scenarios. In these figures, the parameters and initial values used on each case given as follows:

- **Figure 6A ($\mu$ has two negative eigenvalues):**
    - $S_C^\top S_B = \begin{bmatrix} -0.552679 & -0.843293 \\ 0.869146 & -0.967042 \end{bmatrix}$, thus the eigenvalues of $\mu$ are $-0.967445$ and $-0.552276$,
    - $S_\Delta = \begin{bmatrix} 0.287585 & 0.99662 \\ -0.201208 & -0.964587 \end{bmatrix}$,
    - $A = -0.370332 I_2$,
    - initial values $x_1 = (-1.47982, -0.228103)$, $x_2 = (-0.406453, 1.24415)$, $x_3 = (1.8491, -0.625385)$, $x_4 = (1.26989, -1.91216)$.
- **Figure 6B ($\mu$ has one positive eigenvalue and one negative eigenvalue):**
    - $S_C^\top S_B = \begin{bmatrix} -0.155283 & 0.542694 \\ 0.989821 & 0.260748 \end{bmatrix}$, thus the eigenvalues of $\mu$ are $0.846723$ and $-0.741258$,
    - $S_\Delta = \begin{bmatrix} 0.430263 & 0.555071 \\ -0.654555 & 0.422737 \end{bmatrix}$,
    - $A = -0.573698 I_2$,
    - initial values $x_1 = (1.39902, 1.60628)$, $x_2 = (-0.342366, -0.845203)$, $x_3 = (0.616744, 1.63846)$, $x_4 = (-0.185335, -1.34566)$.
- **Figure 6C ($\mu$ has two positive eigenvalues):**
    - $S_C^\top S_B = \begin{bmatrix} 0.981721 & -0.803219 \\ -0.342524 & 0.605171 \end{bmatrix}$, thus the eigenvalues of $\mu$ are $1.39646$ and $0.190429$,
    - $S_\Delta = \begin{bmatrix} 0.653732 & -0.228578 \\ -0.960714 & -0.495344 \end{bmatrix}$,
    - $A = -0.997408 I_2$,
    - initial values $x_1 = (1.1932, -0.702409)$, $x_2 = (-1.49159, -0.735305)$, $x_3 = (1.21287, -0.816296)$, $x_4 = (-0.462258, 1.44549)$.

We observe from Figures 6C that all tokens converge to zero when $\mu$ has only negative eigenvalues. While all tokens diverge to infinity when $\mu$ has at least one positive eigenvalue.

## C  EXPERIMENT DETAILS

In this section, we outline the experimental setup in detail. All experiments are conducted using a server with four A100 GPUs.

### C.1  AUTO-REGRESSIVE LANGUAGE MODELING

**Dataset** We utilize the WIKITEXT103 (Merity et al., 2017), created from Wikipedia articles. It includes a training set of approximately 28,000 articles, amounting to 103 million words in total. Each article is split into sections of about 3,600 words. The validation and test sets contain 60

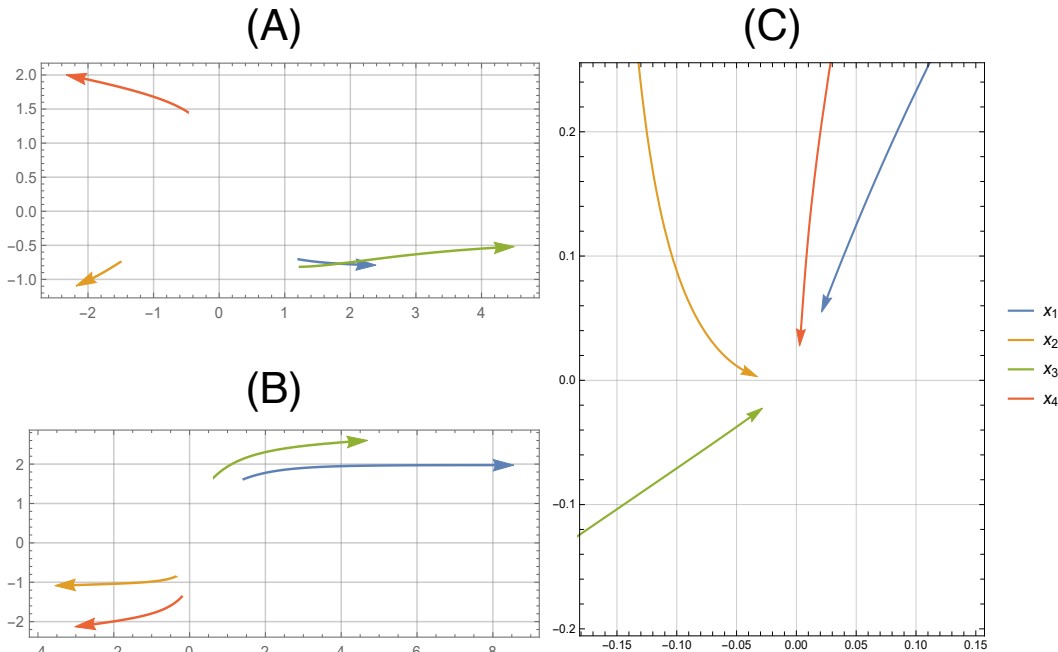

Figure 6: Tokens dynamic behaviors in two-dimensional case where the input/output projection matrix $S_C^\top S_B$ has two positive eigenvalues (A), or one positive eigenvalue (B), or two negative eigenvalues (C). We observe that, in cases (A) and (B), all tokens will diverge to infinity, and in case (C), all tokens will converge to the origin.

articles each, with word counts of 218,000 and 246,000 respectively, combining for a total of roughly 268,000 words.

**Parameterization for three scenarios:** We outline the parameterization used to ensure the S6 layer operates within the three scenarios described in Section 5.1. For the first two scenarios, we leverage the $LDL^T$ decomposition (Golub & Loan, 2013) to ensure the positive or negative definite property of input-output matrix. Specifically, we parameterize $S_C = L^T$ and $S_B = DL^T$, where $D$ is a diagonal matrix and $L$ is a lower triangular matrix with diagonal values fixed at 1. In the first scenario, the diagonal values of $D$ are constrained to be positive and are parameterized as $D = \text{diag}(\text{softplus}(d'))$. In the second scenario, this constraint is reversed by parameterizing $D = -\text{diag}(\text{softplus}(d'))$, ensuring that the diagonal values are negative. In the third scenario, where both positive and negative eigenvalues are required, we modify the diagonal matrix $D$ by changing the sign of half of its diagonal elements. Specifically, we use $D = \text{diag}(s \odot \text{softplus}(d'))$, where $s = [1, \ldots, 1, -1, \ldots, -1]$ is a vector with half of its elements set to 1 and the other half set to $-1$.

**Training details** We train a 130M parameters version of Mamba model (Gu & Dao, 2024) on WIKITEXT103 for 25 epochs with GPT-2 tokenizer (Radford et al., 2019) and AdamW optimizer (Loshchilov & Hutter, 2017). Other hyperparameters are set identically across three scenarios and listed in Table 4.

## C.2 IMAGE CLASSIFICATION

**Dataset** The ImageNet-1K dataset (Deng et al., 2009) is a widely recognized benchmark in the field of computer vision, commonly used for training and evaluating models on the task of large-scale image classification. It contains a collection of 1.28 million labeled training images and 50,000 validation images. Each image is labeled with one of 1,000 distinct classes, representing a broad variety of objects, animals, scenes, and more, allowing the model to learn a rich and diverse range of visual concepts.

Table 4: Hyperparameters configuration for Language Modeling task on WIKITEXT103.

| | |
|---|---|
| Sequence Length | 1024 |
| Peak Learning Rate | 0.0015 |
| Momentum | $\beta_1 = 0.9; \beta_2 = 0.999$ |
| Weight Decay | 0.25 |
| Batch size | 128 |
| Learning rate warmup | Linear |
| Learning rate scheduler | Cosine decay |
| Dropout | 0.25 |

Table 5: Hyperparameters configuration for Image Classification task on ImageNet-1K.

| | |
|---|---|
| Optimizer | AdamW |
| Peak Learning Rate | 0.0008 |
| Momentum | $\beta_1 = 0.9; \beta_2 = 0.999$ |
| Weight Decay | 0.05 |
| Batch size | 512 |
| Learning rate warmup | Linear |
| Learning rate scheduler | Cosine decay |
| Dropout | 0.0 |

**Training details** We train the tiny version (31.8M parameters) of MambaVision model (Hatamizadeh & Kautz, 2024) with and without our token reordering technique. We follow the training procedure in (Yang et al., 2021; Hatamizadeh et al., 2023; Hatamizadeh & Kautz, 2024) and set the hyperparameters for two cases identical, which is listed in Table 5.

## D  ADDITIONAL EXPERIMENTAL RESULTS

### D.1  RE-ORDERING INPUT TOKENS

In this section, we conduct an additional experiment to assess the performance of our re-ordering technique across tasks beyond the vision task reported in Section 5.2.

#### D.1.1  LANGUAGE MODELING

We first present experiment to evaluate the effectiveness of our reordering technique across a broad range of architectures. We use the language modeling task on the WikiText-103 benchmark to test three models: Mamba (Gu & Dao (2024)), H3 (Fu et al. (2023)), and Transformer (Vaswani et al. (2017)). For the latter two models, our importance score in Equation 14 cannot be directly calculated as these models do not have the step size matrix $S_\Delta$. To address this, we made the following adjustments:

- H3 Model: We calculate the token importance score as $s = \langle x, K \rangle$, where $K$ is a learnable vector.
- Transformer Model: We converted a pretrained model (GPT-2 small (Radford et al. (2019)) in our experiment) into the Mamba format and fine-tuned it, following the method outlined in (Wang et al. (2025)). On the converted model, we calculated the token importance score as described in Equation 14.

The results, presented in Table 6, demonstrate that the reordering method consistently improves perplexity across all tested models. These findings highlight the potential of the reordering approach to enhance the performance of a wide range of architectures.

#### D.1.2  TEXT CLASSIFICATION

We now evaluate our method on text classification tasks using the IMDB (Maas et al. (2011)) and AGNews (Zhang et al. (2015)) datasets, using Mamba (Gu & Dao (2024)) as baseline model. The

Table 6: Test perplexity on different models

| Model | With reordering | Without reordering | Improvement with reordering |
|---|---|---|---|
| Mamba | 14.97 | 16.46 | 1.49 |
| H3 | 20.72 | 21.61 | 0.89 |
| Transformer | 13.44 | 16.98 | 3.54 |

results in Table 7 indicate that our re-ordering technique further improves the baseline model's performance on these tasks.

Table 7: Test accuracy ($\uparrow$) on IMDB and AGNews datasets

| Model | IMDB | AGNews |
|---|---|---|
| Mamba (baseline) | 88.46 | 92.08 |
| Mamba + Token reordering | **88.66** | **92.37** |

## D.2 IMPLEMENTATION DETAIL AND COMPUTATIONAL COST

To provide a clearer understanding of our re-ordering technique, we include the pseudo-code for calculating token importance scores and forwarding through a single SSM layer in Algorithm 1. The computational overhead of our technique lies in steps 1, 2, 3, and 4, which require $O(D^2 \times L + D \times L + L \times L + D \times L^2)$ operations per input sequence.

---

**Algorithm 1** Forwarding through a SSM layer with reordering

---

**Require:** Input sequence $\mathbf{x} \in \mathbb{R}^{D \times L}$, hidden matrix $A \in \mathbb{R}^{N \times N}$, input matrix $S_B \in \mathbb{R}^{N \times D}$, output matrix $S_C \in \mathbb{R}^{N \times D}$, step size matrix $S_\Delta \in \mathbb{R}^{D \times D}$, learnable vector $K \in \mathbb{R}^{D \times 1}$, hyperparameters $\tau$ and $p$ of Softsort .
**Ensure:** Output sequence $\mathbf{y} \in \mathbb{R}^{D \times L}$
1: Calculate importance score for each token in sequencce, combined into a vector $\mathbf{s} = (S_\Delta \mathbf{x})^\top K \in \mathbb{R}^{L \times 1}$
2: Sort the score vector in decreasing order: $\mathbf{s}_{sorted} = \text{Sort}(\mathbf{s}) \in \mathbb{R}^{L \times 1}$
3: Calculate the reordering matrix $P = \text{Softmax}(-\frac{(\mathbf{s}_{sorted}\mathbf{1}_L^\top - \mathbf{1}_L\mathbf{s}^\top)^p}{\tau})$ {The power function is applied element-wise and Softmax is applied row-wise}
4: Reorder the input $\mathbf{x}_{\text{reordered}} = \mathbf{x}P^T \in \mathbb{R}^{D \times L}$
5: Calculate the output sequence with selective scan
$\mathbf{y} = \text{SSM}(A, S_B\mathbf{x}_{reordered}, S_C\mathbf{x}_{reordered})(\mathbf{x}_{reordered})$
6: **return** $\mathbf{y}$

---

Additionally, we present the empirical computational costs of our experiment in Section 5.2, benchmarked on the ImageNet classification task using MambaVision-T (Hatamizadeh & Kautz (2024)) as the baseline. Table 8 summarizes key metrics, including memory usage, training time per epoch, parameter count, and FLOPS. The results show that our reordering technique introduces minimal overhead, with only slight increases across all metrics.

## D.3 ROBUSTNESS ON WIKITEXT103 EXPERIMENT

To assess the robustness of our findings across three scenarios of input-output matrices on the WIKITEXT103 (Merity et al. (2017)) language modeling task, we conduct the same experiment described in Section 5.1 with three different random seeds for each scenario and report the mean and standard deviation. The results in Table 9 suggest that our findings are robust to different random seeds.

## D.4 DYNAMIC OF STANDARD MAMBA ARCHITECTURE

In this section, we present an experiment to analyze the token dynamics in the standard Mamba architecture in a practical setting. We evaluate three cases:

Table 8: Comparision of computational cost on model training with MambaVision-T (Hatamizadeh & Kautz (2024)) as Baseline.

| Model | #params | Training time / epoch | Memory / GPU | FLOPS |
|---|---|---|---|---|
| Baseline | 31.79M | 308s | 7232MB | 8.93GFLOPs |
| + reordering | 31.79M | 322s | 7334MB | 8.94GFLOPs |

Table 9: Test perplexity on WIKITEXT103 ($\downarrow$). Mean and standard deviation are computed over three runs with different random seeds. Models with fewer negative eigenvalues in $S_C^\top S_B$ consistently achieve lower perplexity.

| Scenario | Perplexity ($\downarrow$) |
|---|---|
| Negative eigenvalues | $17.28 \pm 0.02$ |
| Mixed eigenvalues | $16.82 \pm 0.02$ |
| Positive eigenvalues | $16.66 \pm 0.05$ |

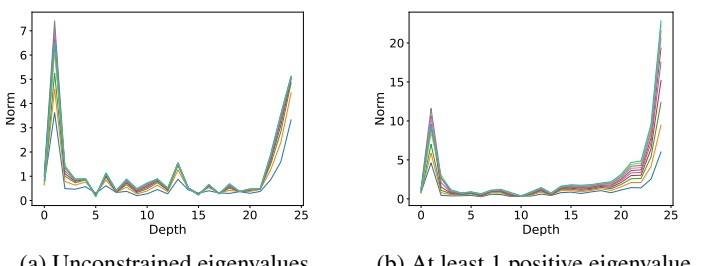 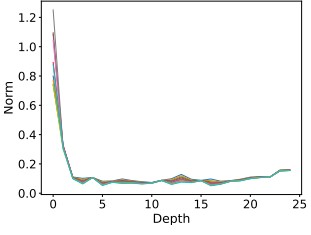

(a) Unconstrained eigenvalues      (b) At least 1 positive eigenvalue      (c) Negative eigenvalues

Figure 7: Token norm evolution in standard Mamba architecture without LayerNorm.

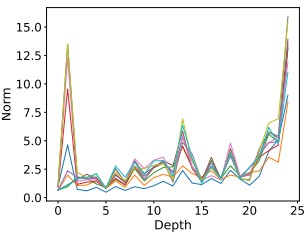 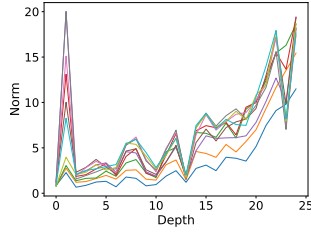 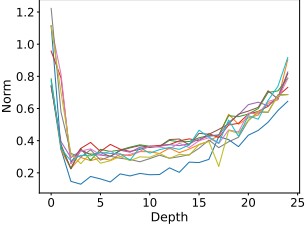

(a) Unconstrained eigenvalues      (b) At least 1 positive eigenvalue      (c) Negative eigenvalues

Figure 8: Token norm evolution in standard Mamba architecture with LayerNorm.

(a) There is no constraint on the input-output projection matrix.

(b) The input-output projection matrix is symmetric and has at least one positive eigenvalue.

(c) The input-output projection matrix is symmetric and has only negative eigenvalues.

We first conduct simulations using a pretrained standard Mamba architecture without LayerNorm on WIKITEXT103. In Figure 7, we visualize the evolution of token norms across the blocks. Figure 7 shows that tokens converge toward zero when the input-output projection matrix has only negative eigenvalues, while tokens diverge when the input-output projection matrix has at least one positive eigenvalue. Moreover, when there is no constraint on the input-output projection matrix, i.e., when we do not force it to always belong to the class of positive definite or negative definite matrices, the tokens' dynamic behaviors become quite complicated. This figure suggests that our theoretical analysis of the tokens' dynamic behaviors in the standard Mamba model without LayerNorm might still be valid.

In addition, we also conduct simulations using a pretrained standard Mamba architecture with LayerNorm in a practical setting on WIKITEXT103. In Figure 8, we visualize the evolution of token norms across the blocks. Figure 8 shows that the tokens' dynamics in a full Mamba setting can be quite complicated. When the input-output projection matrix has only negative eigenvalues,

the tokens do not necessarily tend toward zero. In particular, with $\mathrm{LayerNorm}$ in the Mamba architecture, our theoretical results on the tokens' dynamics might no longer hold.

## D.5 ABLATION STUDY

We perform an ablation study to analyze the effects of various components of Mamba in the token reordering technique. We examine the following configurations, both with and without the reordering technique:

(1) The full Mamba setting,

(2) Mamba setting with weight sharing across layers,

(3) Mamba setting without layer normalization,

(4) Mamba setting without the short convolution layer.

To ensure fairness, all configurations are standardized to approximately 129M parameters. The results, shown in Table 10, highlight the significant contributions of each component and the reordering technique to the model's performance, emphasizing their collective importance in achieving optimal results. Our reordering method consistently yields at least 1.49 PPL improvement in all settings, which is significant for language modeling task with WikiText103.

Table 10: Ablation study on language modeling task with WikiText103.

| Model | With reordering | Without reordering | $\Delta$ |
|---|---|---|---|
| Mamba | **14.97** | 16.46 | 1.49 |
| Mamba + weight sharing | 17.77 | 20.51 | 2.74 |
| Mamba without LayerNorm | 17.82 | 19.31 | 1.49 |
| Mamba without Convolution | 21.81 | 23.38 | 1.57 |

