# OpenReview forum: "Demystifying the Token Dynamics of Deep Selective State Space Models"
_ICLR.cc/2025/Conference — ICLR 2025 Spotlight_

### Official Review · Reviewer_jxmT · 2024-10-21

**Soundness:** 4
**Presentation:** 3
**Contribution:** 2
**Rating:** 8
**Confidence:** 3

**Summary:**

Deep selective state-space models have achieved promising results on sequence modelling tasks. Yet, theoretical understanding of these models remain limited. This paper proposes to study how the embedding of different tokens evolve as they are processed through Mamba layers. In particular, they show that they either converge to 0 or diverge to infinity, based on some criteria that they provide. Importantly, this divergence speed depends on the position of the token. Motivated on their findings, the authors propose some modifications to the standard Mamba architecture that improve its performance.

**Strengths:**

Given the very little theoretical understanding of these models, such a theoretical contribution is valuable to the community. Despite its theoretical focus, the paper is easy to follow and one can grasp the main insights without having to delve into the math. The empirical improvements the theory motivates seem to be significant.

(Score increased from 5 to 8 after the discussion period)

**Weaknesses:**

My main concern with the paper is that its conclusions remain limited compared to what the analysis could, in my opinion, deliver. Here are a few questions that would help, once answered, to improve this point:
- How does the token dynamics in Mamba compare to the ones of Transformers? Are there some qualitative differences? Does it lead to some insights on the different inductive biases of these architectures.
- Is it possible to extend the analysis to the bidirectional case, i.e. without the causal masking (the implicit attention matrix would not be lower triangular anymore)?
- Similar question than the previous point but this time to extend the analysis to gated RNNs such as LSTMs or GRUs, whose simplified versions have some similarity with Mamba.

While it is totally fine to restrain the theoretical analysis to the 1D, simulations assessing whether the conclusions of the analysis on the 1D case also hold in higher dimension are missing. One example of such simulations would be to initialize a Mamba layer in the same way as it is usually done in practice, give it random inputs, and observe how tokens evolve. It might be interesting to play with how correlated the random inputs are and with the magnitude of $a$.

Finally, I remain suspicious of the empirical results given that no learning rate tuning was reported. Without such tuning I find the results hard to trust.

**Questions:**

See above.

---

> ### Author Response · Authors · 2024-11-21
>
> **[Reply 1/2]**
>
> Thank you for your thoughtful review and valuable feedback. Below we address your concerns.
>
> -----
>
> **[W1]-[Q1]. How does the token dynamics in Mamba compare to the ones of Transformers? Are there some qualitative differences? Does it lead to some insights on the different inductive biases of these architectures.**
>
> **Answer:** As outlined in [Geshkovski2024] (page 6, paragraph 2), tokens in a purely self-attention setting typically exhibit exponential divergence to infinity. However, our theoretical analysis demonstrates that the token dynamics in the purely Mamba setting are much more complicated. Specifically, tokens in the Mamba framework may either converge to zero or diverge to infinity. In cases of divergence, the behavior can vary: tokens may diverge rapidly to infinity within a finite time or diverge more gradually to infinity over an infinite time horizon, with the divergence following a logarithmic rate. We have added this discussion to Remark 4 in the revised version of the paper.
>
> **References**
> [Geshkovski2024] Geshkovski, B., Letrouit, C., Polyanskiy, Y., & Rigollet, P. (2024). The emergence of clusters in self-attention dynamics. Advances in Neural Information Processing Systems, 36.
>
> **[Q2]. Is it possible to extend the analysis to the bidirectional case, i.e. without the causal masking (the implicit attention matrix would not be lower triangular anymore)?**
>
> **Answer:** Thanks for your suggestion. We plan to extend our analysis to the bidirectional case in the next few days.
>
> **[Q3]. Similar question than the previous point but this time to extend the analysis to gated RNNs such as LSTMs or GRUs, whose simplified versions have some similarity with Mamba.**
>
> **Answer:** Theorem 1 in [Gu2023] establishes that the classical gating mechanism of RNNs represents a specific instance of the selection mechanism in Mamba. Consequently, our theoretical analysis of Mamba encompasses those RNNs as special cases. A full analysis for a more general gated RNNs such as LSTMs or GRUs would require nontrivial works. We leave this interesting works for future study.
>
> **References**
> [Gu2023] Gu, A., & Dao, T. (2023). Mamba: Linear-time sequence modeling with selective state spaces. arXiv preprint arXiv:2312.00752.
>
> **[W2]. While it is totally fine to restrain the theoretical analysis to the 1D, simulations assessing whether the conclusions of the analysis on the 1D case also hold in higher dimension are missing. One example of such simulations would be to initialize a Mamba layer in the same way as it is usually done in practice, give it random inputs, and observe how tokens evolve. It might be interesting to play with how correlated the random inputs are and with the magnitude of $a$.**
>
> **Answer:** **Regarding conclusions in high-dimentional case:** We conjecture that our theorems still hold in the high-dimensional case. In the high-dimensional case, the input-output matrix $S_C^{\top}S_B$ is a square matrix of size $D \times D$ and it may have complex eigenvalues. In this case, we consider the symmetric matrix $\mu:=\frac{1}{2}(S_C^{\top}S_B)^{\top} + \frac{1}{2}S_C^{\top}S_B$, which is the symmetric part of the input-output projection matrix $S_C^{\top}S_B$. This term is motivated by the observation that: the term $x_l^{\top} \cdot (S_C^{\top}S_B) \cdot x_l$ in the considered dynamical system is a scalar and it can be rewritten as
> $$x_l^{\top} \cdot (S_C^{\top}S_B) \cdot x_l = \frac{1}{2}[x_l^{\top} \cdot (S_C^{\top}S_B) \cdot x_l]^{\top} + \frac{1}{2}[x_l^{\top} \cdot (S_C^{\top}S_B) \cdot x_l] = x_l^{\top} \cdot \mu \cdot x_l.$$
> Since $\mu$ is a symmetric matrix, it always has only real eigenvalues. We particularly conjecture that, in the high-dimensional case, all tokens will converge to zero if $\mu$ has only negative eigenvalues, while all tokens will diverge to infinity if $\mu$ has at least one positive eigenvalue.
>
> To illustrate this conjecture, we run simulations in two-dimensional cases and draw the trajectories of tokens in Figures 6 in Appendix B of the revised version. We observe the following:
>
> - In Figure 6A and 6B, the matrix $\mu$ has at least one positive eigenvalue, and the divergent behavior of the tokens is also observed to persist.
> - In Figure 6C, the matrix $\mu$ has only negative eigenvalues, and the convergent behavior of the tokens is observed to persist.
>
> **Regarding simulations with Mamba setting** involving practical operators (such as layer normalization, linear, or convolution): We plan to provide a few simulations in the next few days to observe the dynamics of tokens.

---

> ### Author Response · Authors · 2024-11-21
>
> **[Reply 2/2]**
>
> **[W3]. Finally, I remain suspicious of the empirical results given that no learning rate tuning was reported. Without such tuning I find the results hard to trust.**
>
> **Answer:** We would like to clarify that our hyperparameter settings follow those established in previous works, specifically Mamba [Gu2023] for language modeling task on WikiText103 and MambaVision [Hatamizadeh2024] for image classification on ImageNet. Additionally, we have provided our code in the supplementary materials, ensuring that our results can be fully reproduced.
>
> **References**
> [Gu2023] Gu, A., & Dao, T. (2023). Mamba: Linear-time sequence modeling with selective state spaces. arXiv preprint arXiv:2312.00752.
> [Hatamizadeh2024] Ali Hatamizadeh and Jan Kautz. Mambavision: A hybrid mamba-transformer vision backbone. arXiv preprint arXiv:2407.08083, 2024.
>
> -----
> We hope we have cleared your concerns about our work. We have also revised our manuscript according to your comments, and we would appreciate it if we can get your further feedback at your earliest convenience.

---

> > ### Comment · Reviewer_jxmT · 2024-11-21
> >
> > Dear authors, thank you for your answer that helps me in better evaluating the paper.
> >
> > There are still two main points I would like to be clarified before updating my score.
> > 1. Regarding the high dimensional case. The 2D simulation is a first step, but I think that we need more. It should be feasible to take a Mamba architecture with standard initialization, no parameter sharing and e.g. 100 layers (without MLPs) and hidden dimension 500 and plot the evolution of the norm of the token over depth. Changing some of the most important parameters of the initialization and observing whether this can change the regime in which the network is would also be particularly insightful. This experiment would also help us understanding whether removing the weight tying assumption breaks the theoretical findings or not.
> > 2. Regarding the learning rate. Do I understand correctly that you take the learning rate that was found to be optimal in previous work and use your method in this setting?

---

> > > ### Author Response · Authors · 2024-11-22
> > >
> > > Thanks for your prompt reply and suggestion. Below we address your additional concerns.
> > >
> > > **Q1. Regarding the high dimensional case. The 2D simulation is a first step, but I think that we need more. It should be feasible to take a Mamba architecture with standard initialization, no parameter sharing [...] and plot the evolution of the norm of the token over depth. [...]**
> > >
> > > **Answer:** Following reviewer's suggestion, we provide additional experiments to analyze the token dynamics in the standard Mamba architecture in a practical setting. We evaluate three cases:
> > >
> > > (i) There is no constraint on the input-output projection matrix.
> > >
> > > (ii) The input-output projection matrix is symmetric and has at least one positive eigenvalue.
> > >
> > > (iii) The input-output projection matrix is symmetric and has only negative eigenvalues.
> > >
> > > We first conduct simulations using a pretrained standard Mamba architecture **without LayerNorm** on WikiText103. In Figure 7 in Appendix D.4 in the revised version, we visualize the evolution of token norms across the blocks. Figure 7 shows that tokens converge toward zero when the input-output projection matrix has only negative eigenvalues, while tokens diverge when the input-output projection matrix has at least one positive eigenvalue. Moreover, when there is no constraint on the input-output projection matrix, i.e., when we do not force it to always belong to the class of positive definite or negative definite matrices on every layer, the tokens' dynamic behaviors become quite complicated. **Figure 7 suggests that our theoretical analysis of the tokens' dynamic behaviors is still valid in the standard Mamba model without LayerNorm.**
> > >
> > >
> > >
> > > In addition, we also conduct simulations using a pretrained standard Mamba architecture with LayerNorm in a practical setting on WikiText103. In Figure 8 in Appendix D.4 in the revised version, we visualize the evolution of token norms across the blocks. Figure 8 shows that the tokens' dynamics in a full Mamba setting can be quite complicated. When the input-output projection matrix has only negative eigenvalues, the tokens do not necessarily tend toward zero. This situation explains the effect of LayerNorm on the tokens' dynamic. Hence, **with LayerNorm in the Mamba architecture, our theoretical results on the tokens' dynamics might no longer hold.**
> > >
> > >
> > > **Q2. Regarding the learning rate. Do I understand correctly that you take the learning rate that was found to be optimal in previous work and use your method in this setting?**
> > >
> > > **Answer:** Yes, we take the learning rate that was found to be optimal in previous work and use it in our setting.
> > >
> > > We have revised our manuscript based on your comments and hope our responses have addressed your concerns about our work. We look forward to receiving your further feedback.

---

> ### Author Response · Authors · 2024-11-23
>
> We would like to thank the reviewer again for your thoughtful reviews and valuable feedback. Beside of our responses above, we would like to provide missing answer on the **possibility of extending the analysis to the bidirectional case** below.
>
> **[Q2] Regarding the possibility of extending the analysis to the bidirectional case:** Our theoretical results on the dynamic behaviors of tokens may no longer hold for bidirectional SSMs (see [Zhu2024] for a definition of bidirectional SSMs). In a bidirectional SSM, input tokens are reordered in two distinct sequences: one in the forward order and the other in the backward order. These two sequences, containing the same tokens but in different arrangements, are processed separately by the SSM before being combined into a single sequence using a summation operator. According to our theoretical results, tokens in earlier positions tend to diverge at a slower rate under the divergence scenario. Consequently, the summation operator mixes token dynamics that diverge at different rates, leading to more complex overall behaviors. We believe that a comprehensive investigation of token dynamics in this bidirectional setting is an interesting direction for future research.
>
> [Zhu2024] Zhu, Lianghui, et al. "Vision mamba: Efficient visual representation learning with bidirectional state space model." arXiv preprint arXiv:2401.09417 (2024).
>
> We would be happy to do any follow-up discussion or address any additional comments.

---

### Official Review · Reviewer_biXN · 2024-10-28

**Soundness:** 2
**Presentation:** 2
**Contribution:** 2
**Rating:** 6
**Confidence:** 4

**Summary:**

The paper provides a theoretical analysis of the dynamical properties of tokens in input-varying SSMs, particularly focusing on the Mamba architecture. The authors derive and analyze the continuous-time limit of the Mamba model, characterizing how tokens evolve through the network. Their key finding is that under several assumptions, in the one-dimensional case, tokens either all converge to zero or all diverge to infinity, with specific criteria determining which scenario occurs. For the divergent case, they prove that different tokens diverge at different rates, suggesting unequal contributions during training. Based on these theoretical insights, they propose two practical improvements: 1. excluding scenarios where tokens converge to zero, as this negatively impacts performance, and 2. reordering tokens based on importance scores. They validate these improvements experimentally on language modeling and image classification tasks.

**Strengths:**

- I believe this is the first paper to present theoretical analysis related to how the token dynamics evolve through the networks across the depth of deep SSMs. This can be valuable to help provide insights into how these models process information
- The mathematical analysis appears to be sound and the empirical results provide some evidence that the theory can lead to insights that can improve performance
- The paper is well-structured and Figures 1-3 are useful.

**Weaknesses:**

- The effects of the assumptions made for the analysis are not thoroughly explored
    - The analysis ignores the effects of important practical additions used in these networks such as layer normalization, short convolutions and the dense linear operations or MLPs often used between layers
    - The analysis assumes weights are shared across layers
    - The analysis assumes the one-dimensional case and excludes the possibility of complex eigenvalues in the input/output projection matrix
    - All of these assumptions are fine for the sake of getting started with a simple case of analysis, but I would have liked to have seen a more thorough discussion of these assumptions and exploration in the empirical results.

- The empirical results seem promising but modest. In addition, it is unclear how much the proposed improvements actually help. In particular, the scenarios investigated empirically seem to have strayed far from the assumptions in the theory (e.g. I believe no weight sharing, higher dimensions, and layer norm, short conv and MLPs included). I would have liked to have seen a series of ablations that build up from the simplified case considered in the theory to the full models that are used in practice. This would give a better sense of how important the assumptions are, and how much the proposed improvements are contributing to the performance boost.

- More care should be taken to distinguish between the depth direction of the network and the sequence direction of the network. SSMs such as S6 come from the prior S4 and S5 methods which are based on continuous-time parameterized ODEs that evolve along the sequence length (often thought of as the "time" direction) and are then discretized. In this work, the continuous-time formulation is taken to be across the depth (i.e. layers) of the network and the "time" is referred to in this depth direction. However, this could lead to much confusion and I would recommend being much more explicit about what the "time" direction is more often throughout the paper to prevent confusion. E.g. in line 212, it is very unclear that time-independent parameters here is meant to mean the parameters are shared across layers and not shared across the sequence length (i.e. a LTI SSM sytem).

**Questions:**

My main concerns I would like addressed are listed in the weaknesses section. Here are some additional comments or questions:

- Perhaps you could start with a model pretty close to that presented in the theory: no layernorm, convs or extra MLPs; weight sharing, 1 dimensional, and then ablate these different choices. How much do each of these choices affect the results? Would also be interesting to see how the results vary with a varying number of layers, e.g. increasing the number of layers to better approximate the "continuous-time limit".

- Wikitext 103 is very much a toy language task that is known to be very sensitive to hyperparameters. How robust are these results to different random seeds? Or different hyperparameters?

- Line 075: The input, output, and state matrices are mislabeled


Update: I have increased my score to reflect my updated view of the paper after the rebuttal period.

---

> ### Author Response · Authors · 2024-11-21
>
> **[Reply 1/2]**
>
> **[W1]. The effects of the assumptions made for the analysis are not thoroughly explored.**
>
> **Answer:** Our primary objective was to examine the most basic form of Mamba that allows for thorough mathematical analysis. We carried out numerical simulations to further validate our tokens dynamic theorems in high-dimensional cases with additional practicle settings in Appendix B of our revised manuscript (with updates highlighted in blue). Interestingly, we found that in general our theorems on tokens dynamic behaviors remained consistent in the high-dimensional case.
>
> **[W2-W6-Q1]. The analysis ignores the effects of important practical additions used in these networks such as layer normalization, short convolutions and the dense linear operations or MLPs often used between layers. [...] Perhaps you could start with a model pretty close to that presented in the theory: no layernorm, convs or extra MLPs; weight sharing, 1 dimensional, and then **ablate these different choices**. How much do each of these choices affect the results?**
>
> **Answer:** Thank you for your suggestion. We plan to extend our experiment in the coming days by including an ablation study on the importance of each layer/component.
>
> **[W3]. The analysis assumes weights are shared across layers.**
>
> **Answer:** In case when the weights are time-dependent, we believe that if the entries of the input-output projection matrix $S_C(t)^\top S_B(t)$, the step size matrix $S_\Delta(t)$, and the scalar $a(t)$ are bounded both above and below, the conclusions of Theorems 4.1, 4.3, and 4.4 remain valid. However, this would require modifications to the proofs to account for these assumptions.
>
> **[W4]. The analysis assumes the one-dimensional case and excludes the possibility of complex eigenvalues in the input/output projection matrix.**
>
> **Answer:** We conjecture that our theorems still hold in the high-dimensional case. In the high-dimensional case, the input-output matrix $S_C^{\top}S_B$ is a square matrix of size $D \times D$ and it may have complex eigenvalues. In this case, we consider the symmetric matrix $\mu:=\frac{1}{2}(S_C^{\top}S_B)^{\top} + \frac{1}{2}S_C^{\top}S_B$, which is the symmetric part of the input-output projection matrix $S_C^{\top}S_B$. This term is motivated by the observation that: the term $x_l^{\top} \cdot (S_C^{\top}S_B) \cdot x_l$ in the considered dynamical system is a scalar and it can be rewritten as
> $$x_l^{\top} \cdot (S_C^{\top}S_B) \cdot x_l = \frac{1}{2}[x_l^{\top} \cdot (S_C^{\top}S_B) \cdot x_l]^{\top} + \frac{1}{2}[x_l^{\top} \cdot (S_C^{\top}S_B) \cdot x_l] = x_l^{\top} \cdot \mu \cdot x_l.$$
> Since $\mu$ is a symmetric matrix, it always has only real eigenvalues. We particularly conjecture that, in the high-dimensional case, all tokens will converge to zero if $\mu$ has only negative eigenvalues, while all tokens will diverge to infinity if $\mu$ has at least one positive eigenvalue.
>
> To illustrate this conjecture, we run simulations in two-dimensional cases and draw the trajectories of tokens in Figures 6 in Appendix B of the revised version. We observe the following:
>
> - In Figure 6A and 6B, the matrix $\mu$ has at least one positive eigenvalue, and the divergent behavior of the tokens is also observed to persist.
> - In Figure 6C, the matrix $\mu$ has only negative eigenvalues, and the convergent behavior of the tokens is observed to persist.
>
> **[W5]. All of these assumptions [one-dim, time-independent weights, no additional operators beside S6 layer, no complex eigenvalues] are fine for the sake of getting started with a simple case of analysis, but I would have liked to have seen a more thorough discussion of these assumptions and exploration in the empirical results.**
>
> **Answer:** We have discussed all of these assumptions in the answers of [W1-W4] above, except an ablation study on the effects of different components/layers which we plan to provide in the next few days. We have also added explorations of these assumptions and additional empirical results in Appendix B (for the high-dimensional case) and Appendix D (for additional experiments) of the revised version of the paper.
>
> In addition to these discussions, we would like to add one more comment on possibly more complex behaviors in the interaction between tokens. After conducting several simulations with random choices for the inputs in the two- and three-dimensional cases, we observe that the dynamic of tokens in purely Mamba setting typically has no periodic, chaotic, nor bifurcation behavior, i.e., if the tokens converge to zero (diverge to infinity), they will converge to zero (diverge to infinity) directly after a few time steps. In particular, the dynamic behavior is quite stable, even when a little noise is added to the initial conditions. We conjecture that the system is stable under a generic assumption of the parameters. A full proof for this conjecture will require adaptations. We leave it for future study.

---

> ### Author Response · Authors · 2024-11-21
>
> **[Reply 2/2]**
>
> **[W7]. More care should be taken to distinguish between the depth direction of the network and the sequence direction of the network. [...]**
>
> **Answer:** Similar to other continuous-time limit of a deep neural network, the time direction in the dynamical system in our paper is exactly the depth direction. Following reviewer's suggestion, we have added an explanation about this time direction below the considered dynamical system in page 4 of the revised version.
>
> **[Q2]. Wikitext 103 is very much a toy language task that is known to be very sensitive to hyperparameters. How robust are these results to different random seeds? Or different hyperparameters?**
>
> **Answer:** To assess the robustness of our findings across three scenarios of input-output matrices on the WikiText103 language modeling task, we conduct the same experiment described in Section 5.1 with three different random seeds for each scenario and report the mean and standard deviation. The results in Table 1 suggest that our findings are robust to different random seeds. We also report these results and findings in Table 9 in Appendix D.3 of our revision.
>
> *Table 1: Test perplexity on WikiText103 ($\downarrow$). Mean and standard deviation are computed over three runs with different random seeds.*
> | **Scenario**             | Perplexity ($\downarrow$) |
> |--------------------------|-------------------|
> | Negative eigenvalues     | 17.28 +- 0.02    |
> | Mixed eigenvalues        | 16.82 +- 0.02    |
> | Positive eigenvalues     | 16.66 +- 0.05    |
>
>
>
> **[Q3]. Line 075: The input, output, and state matrices are mislabeled**
>
> **Answer:** Thank you very much for pointing out this typo. We have updated the sentence. Indeed, the input and output matrices are $S_B$ and $S_C$, respectively.
>
>
> -----
> We hope we have cleared your concerns about our work. We have also revised our manuscript according to your comments, and we would appreciate it if we can get your further feedback at your earliest convenience.

---

> ### Author Response · Authors · 2024-11-22
> **Any Questions from Reviewer biXN on Our Rebuttal?**
>
> We would like to thank the reviewer again for your thoughtful reviews and valuable feedback.
>
> We would appreciate it if you could let us know if our responses have addressed your concerns and whether you still have any other questions about our rebuttal.
>
> We would be happy to do any follow-up discussion or address any additional comments.

---

> > ### Comment · Reviewer_biXN · 2024-11-22
> >
> > I thank the authors' for their detailed response. I will take this into account along with discussions with the reviewers when making the final evaluation and recommendation for the paper.

---

> > > ### Author Response · Authors · 2024-11-22
> > > **Thanks for Your Consideration!**
> > >
> > > Thanks for your prompt reply and consideration. Please let us know if you have any further questions about our submission and rebuttal. We would be happy to engage in follow-up discussions or address any additional comments.
> > >
> > > After discussions with other reviewers, if you agree that our responses to your reviews have addressed the concerns you listed, we kindly ask that you consider whether raising your score would more accurately reflect your updated evaluation of our paper.
> > >
> > > Thank you again for your time and thoughtful comments!

---

> ### Author Response · Authors · 2024-11-23
>
> We would like to thank the reviewer again for your thoughtful reviews and valuable feedback. We would like to provide missing answers on an ablation study to analyze **the effect of practical components on the reordering refinement** below.
>
> **[W2-W6-Q1] Regarding the effects of practical components of Mamba in the token reordering technique:** We conducted an ablation study on WikiText103 and examined the following configurations, both with and without the reordering technique:
>
> (i) the full Mamba model,
>
> (ii) the Mamba with weight sharing across layers,
>
> (iii) the Mamba without layer normalization,
>
> (iv) the Mamba without the short convolution layer.
>
> To ensure fairness, all configurations are standardized to approximately 129M parameters. The results which are shown in Table 4 below highlighting the contribution of each component to the model's performance under our reordering technique.
>
> *Table 4: Ablation study on language modeling task with WikiText103.*
> | Model | With reordering | Without reordering | Improvement with reordering |
> |-|-|-|-|
> | Mamba                        | 14.97           | 16.46             | 1.49       |
> | Mamba + weight sharing       | 17.77           | 20.51             | 2.74       |
> | Mamba without LayerNorm      | 17.82           | 19.31             | 1.49       |
> | Mamba without Convolution    | 21.81           | 23.38             | 1.57       |
>
>
> We have added these results to Appendix D.5 of the revised paper.
>
> We would be happy to do any follow-up discussion or address any additional comments.

---

### Official Review · Reviewer_3W4C · 2024-11-02

**Soundness:** 3
**Presentation:** 3
**Contribution:** 2
**Rating:** 8
**Confidence:** 3

**Summary:**

The paper "Demystifying the Token Dynamics of Deep Selective State Space Models" explores the theoretical behavior of tokens in the Mamba model, a type of Selective State Space Model (SSM) used in sequence tasks like language modeling and image processing. Despite Mamba's empirical success, its internal dynamics are not well understood.
The authors show that in a one-dimensional case, tokens either converge to zero or diverge to infinity, depending on model parameters. Convergence harms performance, while divergence leads to unequal contributions from tokens during training. To address this, the paper proposes two improvements: eliminating the convergent scenario and reordering tokens by importance. Experiments confirm these refinements improve accuracy and speed, particularly in image classification tasks.
This study offers new theoretical insights into token dynamics and suggests practical adjustments to improve the Mamba model's performance.

**Strengths:**

- **Originality**: The paper makes a meaningful contribution by exploring the often overlooked internal dynamics of tokens in Selective State Space Models (SSMs), specifically the Mamba model. While SSMs are well-known for their efficiency in sequence modeling, this work tackles a fresh problem—understanding how token behavior impacts model performance. By analyzing the continuous-time limit of token dynamics, the authors open up a new line of inquiry that could lead to more informed model designs and optimizations. The refinements proposed—eliminating the convergent token scenario and reordering tokens by their importance—are practical ideas that could lead to significant performance gains.
- **Quality**: The paper is technically solid, with rigorous theoretical analysis that provides clear criteria for determining whether tokens in the Mamba model will converge to zero or diverge to infinity. This kind of formal, mathematical foundation is a strong aspect of the paper, ensuring that its claims are well-supported. The empirical results on large-scale tasks such as ImageNet classification offer clear evidence that the proposed refinements are effective in improving model accuracy and convergence speed. The experiments directly tie into the theoretical findings, showcasing a strong link between the analysis and practical outcomes.
- **Clarity**: The paper is clearly structured, with a logical flow from the problem statement, through the theoretical analysis, to the empirical validation. The technical content, while dense in some areas, is well-explained, and the use of diagrams and tables helps to clarify key findings. The authors do a good job of contextualizing their work within existing research, making it clear how their contributions build on prior work in SSMs and neural ODEs. Overall, the clarity of the presentation allows readers to follow the complex theoretical insights without becoming lost in technical details.
- **Significance**: The significance of the work lies in its potential to impact both the theoretical understanding and practical application of SSMs. By uncovering how token dynamics influence model performance, the paper provides a foundation for improving not only Mamba but other models that rely on similar state-space mechanisms. The proposed refinements—removing harmful convergent behaviors and reordering tokens based on importance—offer actionable insights that can directly improve model training efficiency and accuracy, as shown by the experimental results. Given the growing use of SSMs in fields such as language modeling, computer vision, and reinforcement learning, these contributions could lead to broader advancements in how such models are designed and optimized.

**Weaknesses:**

- **Theoretical Scope is Too Narrow**:
While the paper makes a valuable contribution by exploring token dynamics in the one-dimensional case, the focus is too limited. The analysis is restricted to this specific setting, which weakens the generalizability of the conclusions. The authors mention extending their findings to higher-dimensional cases, but without any concrete exploration of these cases, the scope of the theoretical contribution feels incomplete. Actionable Insight: A more comprehensive exploration of higher-dimensional token dynamics should be included. If full proofs are too complex for this paper, at least preliminary results or conjectures, backed by empirical evidence, would make the work stronger. Additionally, comparisons with more sophisticated state-space approaches (such as diagonal vs. non-diagonal state space models) could show how the proposed ideas generalize.
- **Insufficient Diversity of Experiments**:
The empirical validation is focused on tasks like ImageNet classification, which is a strong benchmark, but does not provide sufficient diversity to substantiate the claims made in the paper. The experiments demonstrate the refinements' effectiveness in vision tasks, but the proposed improvements—such as reordering tokens by importance—are presented as potentially broadly applicable to sequence models across various domains. The paper could have benefited from testing the refinements on different types of data and tasks, such as language modeling or reinforcement learning, where SSMs are also commonly applied. Actionable Insight: Expanding the experiments to include tasks like language modeling(e.g., WIKITEXT or other sequence-based datasets) or sequence-based tasks in reinforcement learning would provide stronger evidence that the proposed refinements generalize beyond vision tasks. This would substantiate the claim that these refinements are broadly beneficial across domains.
- **Shallow Investigation of Practical Impact**:
The paper claims that eliminating the convergence scenario and reordering tokens significantly improves the performance of Mamba. While the empirical results show improvements in accuracy and convergence speed, there is little discussion on the practical implications of these refinements in real-world applications. For instance, the computational cost of implementing token reordering or of checking for convergence scenarios during training is not discussed. Additionally, the analysis of why these changes work, especially in larger-scale, practical deployments, is somewhat superficial. Actionable Insight: Including an analysis of the computational overhead or potential trade-offs of implementing these refinements in large-scale models is crucial for practical relevance. Furthermore, deeper explanations of why these adjustments improve performance, with specific references to token dynamics, would enhance the practical significance of the work. For instance, analyzing the impact of token reordering on different architectures or showing how it interacts with other model components would provide more actionable insights for model developers.
- **Limited Novelty of Some Contributions**:
While the paper presents the analysis of token convergence and divergence as a central contribution, this analysis—though useful—builds heavily on well-known concepts from continuous-time neural networks (e.g., neural ODEs) and does not introduce significantly new ideas. The observation that token dynamics can either converge to zero or diverge to infinity, depending on model parameters, is mathematically rigorous but not particularly surprising in the context of dynamical systems. This reduces the novelty of the contribution. Actionable Insight: To strengthen the originality of the work, the authors could extend the analysis beyond what is currently expected in dynamical systems and neural ODEs. For instance, they could explore more complex behaviors in the interaction between tokens, such as the emergence of periodic behavior, chaotic dynamics, or the impact of noise on the stability of the token trajectories. Alternatively, a deeper comparison with transformer-based models (such as interpreting token dynamics in transformers vs. Mamba) would increase the novelty of the work.
- **Lack of Comprehensive Comparison to Prior Work**:
The paper briefly acknowledges prior work on SSMs and transformers but lacks a comprehensive comparison to alternative models. Given the ongoing research in state space models and their applications, a more thorough evaluation against recent models like GSS (Generalized State Spaces) or HiPPO-based models would provide a stronger benchmark for the proposed refinements. Additionally, since the paper suggests that Mamba can be a more efficient alternative to transformers, it should compare the computational efficiency and performance against contemporary transformer models on long-range sequence tasks. Actionable Insight: The authors should include a more detailed experimental comparison against state-of-the-art models in SSMs and transformers. They should also explicitly compare their proposed refinements with other model optimizations, such as low-rank approximations or efficient attention mechanisms in transformers. This would give the community a clearer picture of where Mamba, with its refinements, stands in the current landscape of sequence modeling.

**Questions:**

**Summary of Actionable Improvements**:
- Expand the theoretical analysis to higher dimensions or at least provide empirical evidence for higher-dimensional token dynamics.
- Broaden the experimental validation to include diverse tasks beyond vision (e.g., language modeling and reinforcement learning).
- Discuss the computational costs and trade-offs of the proposed refinements in real-world applications.
- Increase the novelty by exploring more complex dynamical behaviors or offering a deeper comparison with existing models like transformers.
- Provide a more comprehensive experimental comparison to other recent SSMs and transformers, including computational efficiency.

---

> ### Author Response · Authors · 2024-11-21
>
> **[Reply 1/3]**
>
> Thank you for your thoughtful review and valuable feedback. Below we address your concerns.
>
> -----
>
>
>
> **[W1&Q1]. A more comprehensive exploration of higher-dimensional token dynamics should be included. [...] Additionally, comparisons with more sophisticated state-space approaches (such as diagonal vs. non-diagonal state space models) could show how the proposed ideas generalize.**
>
> **Answer:** **Regarding the high-dimensional case:** We conjecture that our theorems still hold in the high-dimensional case. In the high-dimensional case, the input-output matrix $S_C^{\top}S_B$ is a square matrix of size $D \times D$ and it may have complex eigenvalues. In this case, we consider the symmetric matrix $\mu:=\frac{1}{2}(S_C^{\top}S_B)^{\top} + \frac{1}{2}S_C^{\top}S_B$, which is the symmetric part of the input-output projection matrix $S_C^{\top}S_B$. This term is motivated by the observation that: the term $x_l^{\top} \cdot (S_C^{\top}S_B) \cdot x_l$ in the considered dynamical system is a scalar and it can be rewritten as
> $$x_l^{\top} \cdot (S_C^{\top}S_B) \cdot x_l = \frac{1}{2}[x_l^{\top} \cdot (S_C^{\top}S_B) \cdot x_l]^{\top} + \frac{1}{2}[x_l^{\top} \cdot (S_C^{\top}S_B) \cdot x_l] = x_l^{\top} \cdot \mu \cdot x_l.$$
> Since $\mu$ is a symmetric matrix, it always has only real eigenvalues. We particularly conjecture that, in the high-dimensional case, all tokens will converge to zero if $\mu$ has only negative eigenvalues, while all tokens will diverge to infinity if $\mu$ has at least one positive eigenvalue.
>
> To illustrate this conjecture, we run simulations in two-dimensional cases and draw the trajectories of tokens in Figures 6 in Appendix B of the revised version. We observe the following:
>
> - In Figure 6A and 6B, the matrix $\mu$ has at least one positive eigenvalue, and the divergent behavior of the tokens is also observed to persist.
> - In Figure 6C, the matrix $\mu$ has only negative eigenvalues, and the convergent behavior of the tokens is observed to persist.
>
> **Regarding comparisons with more sophisticated state-space approaches:** As described in Algorithms 1 and 2 on page 6 of [Gu2023], S4 and its variants (diagonal, diagonal plus low-rank, HIPPO) can be regarded as special cases of Mamba when the input matrix $B$, output matrix $C$, and step sizes $\Delta$ are input-independent. Consequently, our theoretical analysis of Mamba encompasses those for S4 and its variants as special cases.
>
> **References**
>
> [Gu2023] Gu, A., & Dao, T. (2023). Mamba: Linear-time sequence modeling with selective state spaces. arXiv preprint arXiv:2312.00752.
>
> **[W2&Q2]. Expanding the experiments to include tasks like **language modeling** (e.g., WIKITEXT or other sequence-based datasets) or **sequence-based tasks** in reinforcement learning would provide stronger evidence that the proposed refinements generalize **beyond vision** tasks. This would substantiate the claim that these refinements are broadly beneficial across domains.**
>
> **Answer:** Thanks for your suggestion. We have expanded our experiment to provide evidence that our proposed method can benefits in a more wide range of tasks. In particular, we have benchmarked on two additional tasks, *language modeling* and *text classification*.
>
> For language modeling task, we benchmarked on WikiText103 dataset and achieved perplexity reduction of 1.49, as shown in Table 2.
>
> *Table 2: Test Perplexity ($\downarrow$) on Wikitext-103*
> |**Model**|**Test Perplexity**|
> |-|-|
> |Mamba (baseline)| 16.46|
> |Mamba + reordering| **14.97**|
>
> For text classification, we benchmark on two datasets, IMDB [I] and AGNews [Ag]. The results in Table 3 indicate that our re-ordering technique further improves the baseline model's performance on these tasks.
>
> *Table 3: Test accuracy ($\uparrow$) on IMDB and AGNews datasets*
> |**Model**|**IMDB**| **AGNews**|
> |-|-|-|
> |Mamba (baseline)| 88.46|92.08|
> |Mamba + reordering| **88.66**|**92.37**|
>
> These results suggest that our refinement has the potential to generalize effectively beyond vision-related tasks.

---

> ### Author Response · Authors · 2024-11-21
>
> **[Reply 2/3]**
>
> **[W3&Q3]. Including an **analysis of the computational overhead** or potential trade-offs of implementing these refinements in large-scale models is crucial for practical relevance. Furthermore, deeper explanations of **why these adjustments improve performance, with specific references to token dynamics**, would enhance the practical significance of the work. For instance, **analyzing the impact of token reordering on different architectures** or showing how it **interacts with other model components** would provide more actionable insights for model developers.**
>
>
> **Answer:** Regarding the analysis of computational overhead, we first present the pseudo-code for calculating token importance scores and forwarding through a single SSM layer in Algorithm 1 in Appendix D.2 of our revision and also include this pseudo-code below.
>
>
> #### Algorithm 1: Forwarding through a SSM layer with token reordering
>
> **Input**: Input sequence $\mathbf{x} \in \mathbb{R}^{D \times L}$, hidden matrix $A \in \mathbb{R}^{N \times N}$, input matrix $S_B \in \mathbb{R}^{N \times D}$, output matrix $S_C \in \mathbb{R}^{N \times D}$, step size matrix $S_\Delta \in \mathbb{R}^{D \times D}$, learnable vector $K \in \mathbb{R}^{D \times 1}$, hyperparameters $\tau$ and $p$ of $\operatorname{Softsort}$
>
> **Output**:
> - Output sequence $\mathbf{y} \in \mathbb{R}^{D \times L}$
>
> ---
>
> **Steps**:
>
> 1. **Calculate importance score for every token**: Compute token score vector $\mathbf{s} = (S_\Delta \mathbf{x})^\top K \in \mathbb{R}^{L \times 1}$.
>
> 2. **Sort the score vector in decreasing order**: $\mathbf{s}_{sorted}=\operatorname{Sort}(\mathbf{s}) \in \mathbb{R}^{L\times 1}$
>
> 3. **Calculate the reordering matrix**: $P=\operatorname{Softmax}(-\frac{(\mathbf{s}_{sorted} \mathbb{1}_L^\top-\mathbb{1}_L\mathbf{s}^\top)^p}{\tau}) \in \mathbb{R}^{L \times L}$, where the power function is applied element-wise and $\operatorname{Softmax}$ is applied row-wise.
>
> 4. **Reorder the input**: $\mathbf{x}_{\text{reordered}} = \mathbf{x}P^\top \in \mathbb{R}^{D \times L}$.
>
> 5. **Calculate the output sequence with selective scan:** $$\mathbf{y} = \operatorname{SSM}(A, S_B \mathbf{x}\_{\text{reordered}}, S_C \mathbf{x}\_{\text{reordered}})(\mathbf{x}_{\text{reordered}})$$
>
> 6. **Return** $\mathbf{y}$.
>
> The computational overhead of our technique lies in steps 1, 2, 3, and 4, which require $O(D^2 \times L + D\times L + L\times L + D\times L^2)$ operations per input sequence.
>
>
> Additionally, we present the empirical computational costs of our experiment in Section 5.2, benchmarked on the ImageNet classification task using MambaVision-T [Hatamizadeh2024] as the baseline. Table  4 summarizes key metrics, including memory usage, training time per epoch, parameter count, and FLOPs.
> *Table 4: Comparision of computational cost on model training with MambaVision-T as Baseline.*
> | **Model** | **#params** | **Training time / epoch** | **Training mem / GPU** | **FLOPs** |
> |-|-|-|-|-|
> | MambaVision-T (baseline) | 31.79M | 308s | 7232MB | 8.93GFLOPs |
> | MambaVision-T + reordering | 31.79M | 322s | 7334MB | 8.94GFLOPs |
>
> The results show that our reordering technique introduces minimal overhead, with only slight increases across all metrics.
>
> [Hatamizadeh2024] Ali Hatamizadeh and Jan Kautz. Mambavision: A hybrid mamba-transformer vision backbone. arXiv preprint arXiv:2407.08083, 2024.

---

> ### Author Response · Authors · 2024-11-21
>
> **[Reply 3/3]**
>
> **[W4&Q4]. To strengthen the originality of the work, the authors could extend the analysis beyond what is currently expected in dynamical systems and neural ODEs. For instance, they could **explore more complex behaviors in the interaction** between tokens, such as the emergence of **periodic behavior**, **chaotic dynamics**, or the **impact of noise on the stability of the token trajectories**. Alternatively, a deeper **comparison with transformer-based models** (such as interpreting token dynamics in transformers vs. Mamba) would increase the novelty of the work.**
>
> **Answer:** **Regarding more complex behaviors in the interaction between tokens:** After conducting several simulations with random choices for the inputs in the two- and three-dimensional cases, we observe that the dynamic of tokens in purely Mamba setting typically has no periodic, chaotic, nor bifurcation behavior, i.e., if the tokens converge to zero (diverge to infinity), they will converge to zero (diverge to infinity) directly after a few time steps. In addition, the dynamic behavior is quite stable, even when a little noise is added to the initial conditions. We conjecture that the system is stable under a generic assumption of the parameters. A full proof for this conjecture will require adaptations. We leave it for future study.
>
> **Regarding a comparison of token dynamics with transformer:** As outlined in [Geshkovski2024] (page 6, paragraph 2), tokens in a purely self-attention setting typically exhibit exponential divergence to infinity. However, our theoretical analysis demonstrates that the token dynamics in the purely Mamba setting are much more complicated. Specifically, tokens in the Mamba framework may either converge to zero or diverge to infinity. In cases of divergence, the behavior can vary: tokens may diverge rapidly to infinity within a finite time or diverge more gradually to infinity over an infinite time horizon, with the divergence following a logarithmic rate.
>
> **References**
> [Geshkovski2024] Geshkovski, B., Letrouit, C., Polyanskiy, Y., & Rigollet, P. (2024). The emergence of clusters in self-attention dynamics. Advances in Neural Information Processing Systems, 36.
>
> **[W5&Q5]. The authors should include a more detailed experimental **comparison against state-of-the-art models in SSMs and transformers**. They should also explicitly **compare their proposed refinements with other model optimizations, such as low-rank approximations or efficient attention mechanisms in transformers**. This would give the community a clearer picture of where Mamba, with its refinements, stands in the current landscape of sequence modeling. </font>**
>
> **Answer:** Thank you for your suggestion. We plan to extend our experiment in the coming days by including a more detailed comparison with other SSMs and transformers.
>
>
> -----
> We hope we have cleared your concerns about our work. We have also revised our manuscript according to your comments, and we would appreciate it if we can get your further feedback at your earliest convenience.

---

> ### Author Response · Authors · 2024-11-22
> **Any Questions from Reviewer 3W4C on Our Rebuttal?**
>
> We would like to thank the reviewer again for your thoughtful reviews and valuable feedback.
>
> We would appreciate it if you could let us know if our responses have addressed your concerns and whether you still have any other questions about our rebuttal.
>
> We would be happy to do any follow-up discussion or address any additional comments.

---

> ### Author Response · Authors · 2024-11-24
>
> We would like to thank the reviewer again for your thoughtful reviews and valuable feedback. Beside of our responses above, we would like to provide missing answer on the **effectiveness of our reordering technique across a broad range of architectures** below.
>
> **[W5&Q5]: The authors should include a more detailed experimental comparison against state-of-the-art models in SSMs and transformers [...]**
>
> **Answer:** We conducted an additional experiment to evaluate the effectiveness of our reordering technique across a broad range of architectures. We use the language modeling task on the WikiText-103 benchmark to test three models: Mamba, H3 [1], and Transformer [2]. For the latter two models, no direct equivalent of the step size matrix $S_\Delta$ exists for calculating the token importance score, as defined in Equation (14) of our paper. To adapt the reordering technique to these models, we made the following adjustments:
>
> - **H3 Model**: The token importance score is computed as $s = \langle x, K \rangle$, where $K$ is a learnable vector.
>
> - **Transformer Model**: We converted the pretrained GPT-2 small model into the Mamba format and fine-tuned it, following the method outlined in [3]. On the converted model, we calculated the token importance score as described in Equation (14).
>
> The results, presented in Table 5, demonstrate that our reordering method consistently improves perplexity (PPL) across all tested models. These findings highlight the potential of our approach to enhance the performance of a wide range of architectures.
>
>
> *Table 5: Test perplexity on different models*
> |Name|With reordering| Without reordering|Improvement with reordering |
> |-|-|-|-|
> | Mamba | **14.97**| 16.46 |1.49|
> | H3 | **20.72**|21.61 |0.89|
> | Transformer |**13.44**|16.98|3.54|
>
> We have added these results to Appendix D.6 of the revised paper.
>
> We would be happy to do any follow-up discussion or address any additional comments.
>
> **References**
>
> [1] Fu, Daniel Y., et al. "Hungry hungry hippos: Towards language modeling with state space models." arXiv preprint arXiv:2212.14052 (2022).
>
> [2] Vaswani, A. "Attention is all you need." Advances in Neural Information Processing Systems (2017).
>
> [3] Wang, Junxiong, et al. "The mamba in the llama: Distilling and accelerating hybrid models." arXiv preprint arXiv:2408.15237 (2024).

---

### Official Review · Reviewer_FpJk · 2024-11-04

**Soundness:** 3
**Presentation:** 3
**Contribution:** 3
**Rating:** 8
**Confidence:** 4

**Summary:**

This paper investigates the dynamical properties of tokens in a pre-trained Mamba model, a type of selective state space model (SSM). Despite the empirical success of Mamba, a comprehensive theoretical understanding of its internal mechanisms is lacking in various areas. The authors address this gap in one area by deriving the dynamical system governing the continuous-time limit of the Mamba model and analyzing its asymptotic behavior. In the one-dimensional case, they prove that the system exhibits one of two behaviors: either all tokens converge to zero or diverge to infinity. They provide criteria based on model parameters to determine which scenario occurs. Empirically, they verify that the convergent scenario negatively impacts model performance, while in the divergent scenario, tokens diverge at different rates, contributing unequally during training.
Based on these findings, the authors propose two refinements to improve model performance:
* Excluding the Convergent Scenario: By ensuring certain conditions on model parameters, they prevent the tokens from converging to zero.
* Reordering Tokens: They reorder tokens based on their importance scores to account for the unequal contribution during training.

Experimental results on language modeling and image classification tasks validate these refinements, showing improved accuracy and convergence speed.

**Strengths:**

- Originality: The paper addresses a gap in the theoretical understanding of selective state space models, specifically Mamba, by analyzing the continuous-time dynamics of tokens.
- Theoretical Rigor: Provides rigorous mathematical proofs for the asymptotic behavior of tokens, offering clear criteria based on model parameters.
- Practical Implications: The findings lead to actionable refinements that improve model performance, validated through experiments on language modeling and image classification tasks.
- Clarity: The paper is well-structured, with logical progression and clear explanations that make complex mathematical concepts accessible.

**Weaknesses:**

- Simplifying Assumptions: Using time-independent parameters and excluding other layers for mathematical convenience may oversimplify Mamba's actual behavior in practical settings.
- Experimental Scope: Though supportive, experimental validation could be more extensive. Including additional datasets and comparisons with other models would strengthen the claims.
- Limited Discussion on Limitations: The paper could benefit from a more thorough discussion of the limitations of the proposed refinements and potential challenges in practical implementation.

**Questions:**

1. Could you extend the theoretical analysis to higher-dimensional cases?
2. Can you provide more details on how the token importance scores are computed and how the reordering is implemented? What is the computational overhead of this process?
3. Do your findings and proposed refinements generalize to other selective state space models beyond Mamba?

---

> ### Author Response · Authors · 2024-11-21
>
> **[Reply 1/3]**
>
> Thank you for your thoughtful review and valuable feedback. Below we address your concerns.
>
> -----
>
>
> **[W1]. Simplifying Assumptions: Using time-independent parameters and excluding other layers for mathematical convenience may oversimplify Mamba's actual behavior in practical settings.**
>
> **Answer:** Our primary objective was to examine the most basic form of Mamba that allows for thorough mathematical analysis. We carried out a few numerical simulations to further validate our tokens dynamic theorems in **high-dimensional cases** in the revised version (with updates highlighted in blue). Interestingly, we found that in general our theorems on tokens dynamic behaviors remained consistent in the high-dimensional case.
>
> Regarding the **time-indepedent weights assumption**, we believe that if the entries of the input/output projection matrix $S_C(t)^\top S_B(t)$, the step size matrix $S_\Delta(t)$, and the scalar $a(t)$ are bounded both above and below, the conclusions of Theorems 4.1, 4.3, and 4.4 remain valid. However, this would require modifications to the proofs to account for these assumptions.
>
> Regarding a more complicated setting with additional practical layers, we will provide a few simulation results in the next few days to illustrate the possibility of generalizing our theorems in these cases.
>
> **[W2]. **Experimental Scope**: Though supportive, experimental validation could be more extensive. Including additional datasets and comparisons with other models would strengthen the claims.**
>
> **Answer:** Following your suggestion, we have conducted additional experiments to better assess the performance of our proposed method. Specifically, we benchmarked on two additional tasks: *language modeling* and *text classification*. We also plan to conduct an experiment to with other models (SSMs and Transformers) in the next few days.
>
> For the language modeling task, we evaluated our method on the WikiText-103 dataset and achieved a perplexity reduction of 1.49, as shown in Table 2.
>
>
> *Table 2: Test Perplexity ($\downarrow$) on WikiText103*
> |**Model**|**Test Perplexity**|
> |-|-|
> |Mamba (baseline)| 16.46|
> |Mamba + reordering| **14.97**|
>
> For text classification, we benchmark on two datasets, IMDB [Maas2011] and AGNews [Zhang2015]. The results in Table 3 indicate that our re-ordering technique further improves the baseline model's performance on these tasks.
>
> *Table 3: Test accuracy ($\uparrow$) on IMDB and AGNews datasets*
> |**Model**|**IMDB**| **AGNews**|
> |-|-|-|
> |Mamba (baseline)| 88.46|92.08|
> |Mamba + reordering| **88.66**|**92.37**|
>
>
> **Reference**
> [Maas2011] Andrew L. Maas, Raymond E. Daly, Peter T. Pham, Dan Huang, Andrew Y. Ng, and Christopher Potts. Learning word vectors for sentiment analysis. Proceedings of the 49th Annual Meeting of the Association for Computational Linguistics: Human Language Technologies, 2011. URL https://aclanthology.org/P11-1015.
>
> [Zhang2015] Xiang Zhang, Junbo Zhao, and Yann LeCun. Character-level convolutional networks for text classification. Advances in Neural Information Processing Systems, 2015. https://proceedings.neurips.cc/paper_files/paper/2015/file/250cf8b51c773f3f8dc8b4be867a9a02-Paper.pdf.
>
> **[W3]. Limited Discussion on Limitations: The paper could benefit from a more thorough discussion of the limitations of the proposed refinements and potential challenges in practical implementation**
>
> **Answer:** Following your suggestion, we have added discussions on the limitations of the proposed refinements and potential challenges in practical implementation to the limitation part of the conlusion section in the revised version. In particular, our token reordering refinement, based on importance scores, introduces additional computational overhead during training, although this increase is relatively small. Furthermore, we are uncertain whether the method of reordering tokens based on their orthogonal projection onto a learnable affine line, as used in practical implementation, is optimal. We leave a systematic study of token reordering refinements for future work.

---

> ### Author Response · Authors · 2024-11-21
>
> **[Reply 2/3]**
>
> **[Q1]. Could you extend the theoretical analysis to higher-dimensional cases?**
>
> **Answer:** We conjecture that our theorems still hold in the high-dimensional case. A full proof for this conjecture will require a nontrivial work. In the high-dimensional case, the input-output matrix $S_C^{\top}S_B$ is a square matrix of size $D \times D$, and it may have complex eigenvalues. In this case, we consider the symmetric matrix $\mu:=\frac{1}{2}(S_C^{\top}S_B)^{\top} + \frac{1}{2}S_C^{\top}S_B$, which is the symmetric part of the input-output projection matrix $S_C^{\top}S_B$. This term is motivated by the observation that the term $x_l^{\top} \cdot (S_C^{\top}S_B) \cdot x_l$ in the considered dynamical system is a scalar and it can be rewritten as
> $$x_l^{\top} \cdot (S_C^{\top}S_B) \cdot x_l = \frac{1}{2}[x_l^{\top} \cdot (S_C^{\top}S_B) \cdot x_l]^{\top} + \frac{1}{2}[x_l^{\top} \cdot (S_C^{\top}S_B) \cdot x_l] = x_l^{\top} \cdot \mu \cdot x_l.$$
> Since $\mu$ is a symmetric matrix, it always has only real eigenvalues. We particularly conjecture that, in the high-dimensional case, all tokens will converge to zero if $\mu$ has only negative eigenvalues, while all tokens will diverge to infinity if $\mu$ has at least one positive eigenvalue.
>
> To illustrate this conjecture, we run simulations in two-dimensional cases and draw the trajectories of tokens in Figures 6 in Appendix B of the revised manuscript. We observe the following:
>
>
> - In Figure 6A and 6B, the matrix $\mu$ has at least one positive eigenvalue, and the divergent behavior of the tokens is also observed to persist.
> - In Figure 6C, the matrix $\mu$ has only negative eigenvalues, and the convergent behavior of the tokens is observed to persist.
>
> We have added these discussions with further details in Appendix B.
>
> **[Q2].** Can you provide more details on how the token importance scores are computed and how the reordering is implemented? What is the computational overhead of this process?
>
> **Answer.** To provide a clearer understanding of our re-ordering technique, we present the pseudo-code for calculating token importance scores and forwarding through a single SSM layer in Algorithm 1 in Appendix D.2 of our revision and also include this pseudo-code below.
>
>
>
> #### Algorithm 1: Forwarding through a SSM layer with token reordering
>
> **Input**: Input sequence $\mathbf{x} \in \mathbb{R}^{D \times L}$, hidden matrix $A \in \mathbb{R}^{N \times N}$, input matrix $S_B \in \mathbb{R}^{N \times D}$, output matrix $S_C \in \mathbb{R}^{N \times D}$, step size matrix $S_\Delta \in \mathbb{R}^{D \times D}$, learnable vector $K \in \mathbb{R}^{D \times 1}$, hyperparameters $\tau$ and $p$ of $\operatorname{Softsort}$
>
> **Output**:
> - Output sequence $\mathbf{y} \in \mathbb{R}^{D \times L}$
>
> ---
>
> **Steps**:
>
> 1. **Calculate importance score for every token**: Compute token score vector $\mathbf{s} = (S_\Delta \mathbf{x})^\top K \in \mathbb{R}^{L \times 1}$.
>
> 2. **Sort the score vector in decreasing order**: $\mathbf{s}_{sorted}=\operatorname{Sort}(\mathbf{s}) \in \mathbb{R}^{L\times 1}$
>
> 3. **Calculate the reordering matrix**: $P=\operatorname{Softmax}(-\frac{(\mathbf{s}_{sorted} \mathbb{1}_L^\top-\mathbb{1}_L\mathbf{s}^\top)^p}{\tau}) \in \mathbb{R}^{L \times L}$, where the power function is applied element-wise and $\operatorname{Softmax}$ is applied row-wise.
>
> 4. **Reorder the input**: $\mathbf{x}_{\text{reordered}} = \mathbf{x}P^\top \in \mathbb{R}^{D \times L}$.
>
> 5. **Calculate the output sequence with selective scan:** $$\mathbf{y} = \operatorname{SSM}(A, S_B \mathbf{x}\_{\text{reordered}}, S_C \mathbf{x}\_{\text{reordered}})(\mathbf{x}\_{\text{reordered}})$$
>
> 6. **Return** $\mathbf{y}$.
>
> The computational overhead of our technique lies in steps 1, 2, 3, and 4, which require $O(D^2 \times L + D\times L + L\times L + D\times L^2)$ operations per input sequence.
>
>
> Additionally, we present the empirical computational costs of our experiment in Section 5.2, benchmarked on the ImageNet classification task using MambaVision-T [Hatamizadeh2024] as the baseline. Table  4 summarizes key metrics, including memory usage, training time per epoch, parameter count, and FLOPs.
> *Table 4: Comparision of computational cost on model training with MambaVision-T as Baseline.*
> | **Model** | **#params** | **Training time / epoch** | **Training mem / GPU** | **FLOPs** |
> |-|-|-|-|-|
> | MambaVision-T (baseline) | 31.79M | 308s | 7232MB | 8.93GFLOPs |
> | MambaVision-T + reordering | 31.79M | 322s | 7334MB | 8.94GFLOPs |
>
> The results show that our reordering technique introduces minimal overhead, with only slight increases across all metrics.
>
> [Hatamizadeh2024] Ali Hatamizadeh and Jan Kautz. Mambavision: A hybrid mamba-transformer vision backbone. arXiv preprint arXiv:2407.08083, 2024.

---

> ### Author Response · Authors · 2024-11-21
>
> **[Reply 3/3]**
>
> **[Q3]. Do your findings and proposed refinements generalize to other selective state space models beyond Mamba?**
>
> **Answer:** Yes, our findings and proposed refinements generalize to other selective state space models beyond Mamba. Indeed, as described in Algorithms 1 and 2 on page 6 of [Gu2023], S4 and its variants (diagonal, diagonal plus low-rank, HIPPO) can be regarded as special cases of Mamba when the input matrix $B$, output matrix $C$, and step sizes $\Delta$ are input-independent and are chosen carefully. Consequently, our theoretical analysis and the proposed refinements of Mamba encompasses those for S4 and its variants as special cases.
>
> **References**
> [Gu2023] Gu, A., & Dao, T. (2023). Mamba: Linear-time sequence modeling with selective state spaces. arXiv preprint arXiv:2312.00752.
>
> -----
> We hope we have cleared your concerns about our work. We have also revised our manuscript according to your comments, and we would appreciate it if we can get your further feedback at your earliest convenience.

---

> ### Author Response · Authors · 2024-11-22
> **Any Questions from Reviewer FpJk on Our Rebuttal?**
>
> We would like to thank the reviewer again for your thoughtful reviews and valuable feedback.
>
> We would appreciate it if you could let us know if our responses have addressed your concerns and whether you still have any other questions about our rebuttal.
>
> We would be happy to do any follow-up discussion or address any additional comments.

---

> ### Author Response · Authors · 2024-11-24
>
> We would like to thank the reviewer again for your thoughtful reviews and valuable feedback. Beside of our responses above, we would like to provide missing answers on the **actual behavior of tokens' dynamic in practical setting** and **effectiveness of our reordering technique across a broad range of architectures** below.
>
> **[W1]. [...] Mamba's actual behavior in practical settings:** Following reviewers' suggestion, we provide additional experiments to analyze the token dynamics in the standard Mamba architecture in a practical setting. We evaluate three cases:
>
> (i) There is no constraint on the input-output projection matrix.
>
> (ii) The input-output projection matrix is symmetric and has at least one positive eigenvalue.
>
> (iii) The input-output projection matrix is symmetric and has only negative eigenvalues.
>
> We first conduct simulations using a pretrained standard Mamba architecture without LayerNorm on WikiText103. In Figure 7 in Appendix D.4 in the revised version, we visualize the evolution of token norms across the blocks. Figure 7 shows that tokens converge toward zero when the input-output projection matrix has only negative eigenvalues, while tokens diverge when the input-output projection matrix has at least one positive eigenvalue. Moreover, when there is no constraint on the input-output projection matrix, i.e., when we do not force it to always belong to the class of positive definite or negative definite matrices on every layer, the tokens' dynamic behaviors become quite complicated. **Figure 7 suggests that our theoretical analysis of the tokens' dynamic behaviors is still valid in the standard Mamba model without LayerNorm.**
>
> In addition, we also conduct simulations using a pretrained standard Mamba architecture with LayerNorm in a practical setting on WikiText103. In Figure 8 in Appendix D.4 in the revised version, we visualize the evolution of token norms across the blocks. Figure 8 shows that the tokens' dynamics in a full Mamba setting can be quite complicated. When the input-output projection matrix has only negative eigenvalues, the tokens do not necessarily tend toward zero. This situation explains the effect of LayerNorm on the tokens' dynamic. Hence, **with LayerNorm in the Mamba architecture, our theoretical results on the tokens' dynamics might no longer hold.**
>
> **[W2]: [...] Including [...] comparisons with other models would strengthen the claims.**
>
> **Answer:** We have conducted an additional experiment to evaluate the effectiveness of our reordering technique across a broad range of architectures. We use the language modeling task on the WikiText-103 benchmark to test three models: Mamba, H3 [1], and Transformer [2]. For the latter two models, no direct equivalent of the step size matrix $S_\Delta$ exists for calculating the token importance score, as defined in Equation (14) of our paper. To adapt the reordering technique to these models, we made the following adjustments:
>
> - **H3 Model**: The token importance score is computed as $s = \langle x, K \rangle$, where $K$ is a learnable vector.
>
> - **Transformer Model**: We converted the pretrained GPT-2 small model into the Mamba format and fine-tuned it, following the method outlined in [3]. On the converted model, we calculated the token importance score as described in Equation (14).
>
> The results, presented in Table 5, demonstrate that our reordering method consistently improves perplexity (PPL) across all tested models. These findings highlight the potential of our approach to enhance the performance of a wide range of architectures.
>
>
> *Table 5: Test perplexity on different models*
> |Name|With reordering| Without reordering|Improvement with reordering |
> |-|-|-|-|
> | Mamba | **14.97**| 16.46 |1.49|
> | H3 | **20.72**|21.61 |0.89|
> | Transformer |**13.44**|16.98|3.54|
>
> We have added these results to Appendix D.6 of the revised paper.
>
> We would be happy to do any follow-up discussion or address any additional comments.
>
> **References**
>
> [1] Fu, Daniel Y., et al. "Hungry hungry hippos: Towards language modeling with state space models." arXiv preprint arXiv:2212.14052 (2022).
>
> [2] Vaswani, A. "Attention is all you need." Advances in Neural Information Processing Systems (2017).
>
> [3] Wang, Junxiong, et al. "The mamba in the llama: Distilling and accelerating hybrid models." arXiv preprint arXiv:2408.15237 (2024).

---

### Author Response · Authors · 2024-11-21
**Additional Experiments**

**1. Regarding the performance of token reordering technique on additional tasks:** We have conducted additional experiments to better assess the performance of our proposed method. Specifically, we benchmarked on two additional tasks: *language modeling* and *text classification*.

For the language modeling task, we evaluated our method on the WikiText-103 dataset and achieved a perplexity reduction of 1.49, as shown in Table 2.


*Table 2: Test Perplexity ($\downarrow$) on WikiText103*
|**Model**|**Test Perplexity**|
|-|-|
|Mamba (baseline)| 16.46|
|Mamba + reordering| **14.97**|

For text classification, we benchmark on two datasets, IMDB [Maas2011] and AGNews [Zhang2015]. The results in Table 3 indicate that our re-ordering technique further improves the baseline model's performance on these tasks.

*Table 3: Test accuracy ($\uparrow$) on IMDB and AGNews datasets*
|**Model**|**IMDB**| **AGNews**|
|-|-|-|
|Mamba (baseline)| 88.46|92.08|
|Mamba + reordering| **88.66**|**92.37**|

These results suggest that our refinement has the potential to generalize effectively beyond vision-related tasks.

**Reference**
[Maas2011] Andrew L. Maas, Raymond E. Daly, Peter T. Pham, Dan Huang, Andrew Y. Ng, and Christopher Potts. Learning word vectors for sentiment analysis. Proceedings of the 49th Annual Meeting of the Association for Computational Linguistics: Human Language Technologies, 2011. URL https://aclanthology.org/P11-1015.

[Zhang2015] Xiang Zhang, Junbo Zhao, and Yann LeCun. Character-level convolutional networks for text classification. Advances in Neural Information Processing Systems, 2015. https://proceedings.neurips.cc/paper_files/paper/2015/file/250cf8b51c773f3f8dc8b4be867a9a02-Paper.pdf.



**2. Details regarding the implementation of reordering and its computational efficiency:**

To provide a clearer understanding of our re-ordering technique, we present the pseudo-code for calculating token importance scores and forwarding through a single SSM layer in Algorithm 1 in Appendix D of the revised version of our paper. The computational overhead of our algorithm is $O(D^2 \times L + D\times L + L\times L + D\times L^2)$ operations per input sequence.


Additionally, we present the empirical computational costs of our experiment in Section 5.2, benchmarked on the ImageNet classification task using MambaVision-T [Hatamizadeh2024] as the baseline. Table  4 summarizes key metrics, including memory usage, training time per epoch, parameter count, and FLOPs.

*Table 4: Comparision of computational cost on model training with MambaVision-T as Baseline.*
| **Model** | **#params** | **Training time / epoch** | **Training mem / GPU** | **FLOPs** |
|-|-|-|-|-|
| MambaVision-T (baseline) | 31.79M | 308s | 7232MB | 8.93GFLOPs |
| MambaVision-T + reordering | 31.79M | 322s | 7334MB | 8.94GFLOPs |

The results show that our reordering technique introduces minimal overhead, with only slight increases across all metrics.

[Hatamizadeh2024] Ali Hatamizadeh and Jan Kautz. Mambavision: A hybrid mamba-transformer vision backbone. arXiv preprint arXiv:2407.08083, 2024.

**3. Regarding the robustness of our method on WikiText-103 Experiment**:
To assess the robustness of our findings across three scenarios of input-output matrices on the WikiText103 language modeling task, we conduct the same experiment described in Section 5.1 with three different random seeds for each scenario and report the mean and standard deviation. The results in Table 1 suggest that our findings are robust to different random seeds. We also report these results and findings in Table 9 in Appendix D.3 of our revision.

*Table 1: Test perplexity on WikiText103 ($\downarrow$). Mean and standard deviation are computed over three runs with different random seeds.*
| **Scenario**             | Perplexity ($\downarrow$) |
|--------------------------|-------------------|
| Negative eigenvalues     | 17.28 +- 0.02    |
| Mixed eigenvalues        | 16.82 +- 0.02    |
| Positive eigenvalues     | 16.66 +- 0.05    |

---

### Author Response · Authors · 2024-11-21
**General Response**

Dear AC and Reviewers,

Thanks for your thoughtful reviews and valuable comments, which have helped us improve the paper significantly. We are encouraged by the endorsements that: 1) Our paper is the first to present a theoretical analysis of how token dynamics evolve through deep SSMs (Reviewer biXN), addresses a gap in the theoretical understanding of SSMs (Reviewers FpJk and jxmT), and opens up a new line of inquiry that could lead to more informed model designs and optimizations (Reviewer 3W4C); and 2) The refinements proposed are practical ideas that could lead to significant performance gains, contributing to broader advancements in how such models are designed and optimized (Reviewer 3W4C).

All reviewers' comments/suggestions have been addresses and updated in the revised version of the paper (except a few simulations that we will report in the next few days). In the following, we address some of the common comments from Reviewers.


**1. Regarding a generalization of our theorems to high-dimension:** We conjecture that our theorems still hold in the high-dimensional case. In the high-dimensional case, the input-output matrix $S_C^{\top}S_B$ is a square matrix of size $D \times D$ and it may have complex eigenvalues. In this case, we consider the symmetric matrix $\mu:=\frac{1}{2}(S_C^{\top}S_B)^{\top} + \frac{1}{2}S_C^{\top}S_B$, which is the symmetric part of the input-output projection matrix $S_C^{\top}S_B$. This term is motivated by the observation that: the term $x_l^{\top} \cdot (S_C^{\top}S_B) \cdot x_l$ in the considered dynamical system is a scalar and it can be rewritten as
$$x_l^{\top} \cdot (S_C^{\top}S_B) \cdot x_l = \frac{1}{2}[x_l^{\top} \cdot (S_C^{\top}S_B) \cdot x_l]^{\top} + \frac{1}{2}[x_l^{\top} \cdot (S_C^{\top}S_B) \cdot x_l] = x_l^{\top} \cdot \mu \cdot x_l.$$
Since $\mu$ is a symmetric matrix, it always has only real eigenvalues. We particularly conjecture that, in the high-dimensional case, all tokens will converge to zero if $\mu$ has only negative eigenvalues, while all tokens will diverge to infinity if $\mu$ has at least one positive eigenvalue.

To illustrate this conjecture, we run simulations in two-dimensional cases and draw the trajectories of tokens in Figures 6 in Appendix B of the revised version. We observe the following:

- In Figure 6A and 6B, the matrix $\mu$ has at least one positive eigenvalue, and the divergent behavior of the tokens is also observed to persist.
- In Figure 6C, the matrix $\mu$ has only negative eigenvalues, and the convergent behavior of the tokens is observed to persist.

**2. Regarding a comparison to the tokens dynamic of Transformer:** As outlined in [A] (page 6, paragraph 2), tokens in a purely self-attention setting typically exhibit exponential divergence to infinity. However, our theoretical analysis demonstrates that the token dynamics in the purely Mamba setting are much more complicated. Specifically, tokens in the Mamba framework may either converge to zero or diverge to infinity. In cases of divergence, the behavior can vary: tokens may diverge rapidly to infinity within a finite time or diverge more gradually to infinity over an infinite time horizon, with the divergence following a logarithmic rate.

[A] Geshkovski, B., Letrouit, C., Polyanskiy, Y., & Rigollet, P. (2024). The emergence of clusters in self-attention dynamics. Advances in Neural Information Processing Systems, 36.

In addition to addressing the methodology, and following the reviewers' suggestions, we have conducted **additional simulations and experiments** to examine our refinements (see **Additional Experiments** below). We are glad to answer any further questions you have on our submission.

---

### Author Response · Authors · 2024-11-21
**Summary of Revisions**

Incorporating comments and suggestions from reviewers, as well as some further informative empirical studies, we summarize here the main changes in the revised paper:

1. We have added further discussion and simulations for high-dimensional cases in Remark 3 in the main text and in Appendix B. We conjecture that our main theorems on the tokens' dynamic behaviors still hold in the high-dimensional case. We leave the study of the higher-dimensional case for future work.
2. We have added comparison on the tokens' dynamic behavior of Mamba and Transformer in Remark 4 in the main text. Our theoretical analysis demonstrates that the token dynamics in the purely Mamba setting are much more complicated than those in the purely self-attention setting.
3. We have also added discussions on limitations of our refinement methods inthe limitation part of the conclusion section. Our token reordering refinement introduces additional computational overhead during training, although this increase is relatively small. Furthermore, we are uncertain whether the method of reordering tokens used in our setting is optimal.
4. We conduct an additional experiment in Appendix D.1 to assess the performance of our re-ordering technique across tasks beyond the vision task. The additional tasks are  language modeling task using the WIKITEXT103 datase and text classification tasks using the IMDB and AGNews datasets. Experimental results show that our re-ordering technique further improves the baseline model’s performance on these tasks.
5. We include the pseudo-code for calculating token importance scores and forwarding through a single SSM layer in Algorithm 1 and determine the computational overhead of our technique in Appendix D.2.
6. We further conduct the experiment described in Section 5.1 with three different random seeds for our three scenarios to assess the robustness of our findings across three scenarios of input-output matrices on the WIKI-TEXT103 language modeling task. The results is reported in Appendix D.3 and they suggest that our findings are robust to different random seeds.
7. We also added an experiment in Appendix D.4 to analyze the token dynamics in the standard Mamba architecture in a practical setting. The experimental results suggest that our theoretical analysis of the tokens' dynamic behaviors in the standard Mamba model without LayerNorm might still be valid. However, with LayerNorm in the Mamba architecture, our theoretical results on the tokens' dynamics might no longer hold.
8. In addition, in Appendix D.5, we perform an ablation study to analyze the effects of various components of Mamba in the token reordering technique. The results highlight the significant contributions of each component and the reordering technique to the model's performance, emphasizing their collective importance in achieving optimal results. Our reordering method consistently yields at least 1.49 PPL improvement in all settings, which is significant for language modeling task with WikiText103.
9. Finally, we also present experiment in Appendix D.6 to evaluate the effectiveness of our reordering technique across a broad range of architectures. We use the language modeling task on the WikiText-103 benchmark to test three models: Mamba, H3, and Transformer. The results demonstrate that the reordering method consistently improves perplexity across all tested models.

---

### Author Response · Authors · 2024-11-23
**General Responses 2**

Dear AC and Reviewers,

We would like to thank the reviewer again for your thoughtful reviews and valuable feedback. We would like to provide missing answers on: **(1) tokens' dynamic behaviors in the practical implementation of Mamba**, and **(2) the effect of practical components on the reordering refinement**, and **(3) possibility of extending the analysis to the bidirectional case** below.

**(1) Regarding the tokens' dynamic behaviors in Mamma setting used in practical implementation:** Following reviewers' suggestion, we provide additional experiments to analyze the token dynamics in the standard Mamba architecture in a practical setting. We evaluate three cases:

(i) There is no constraint on the input-output projection matrix.

(ii) The input-output projection matrix is symmetric and has at least one positive eigenvalue.

(iii) The input-output projection matrix is symmetric and has only negative eigenvalues.

We first conduct simulations using a pretrained standard Mamba architecture without LayerNorm on WikiText103. In Figure 7 in Appendix D.4 in the revised version, we visualize the evolution of token norms across the blocks. Figure 7 shows that tokens converge toward zero when the input-output projection matrix has only negative eigenvalues, while tokens diverge when the input-output projection matrix has at least one positive eigenvalue. Moreover, when there is no constraint on the input-output projection matrix, i.e., when we do not force it to always belong to the class of positive definite or negative definite matrices on every layer, the tokens' dynamic behaviors become quite complicated. **Figure 7 suggests that our theoretical analysis of the tokens' dynamic behaviors is still valid in the standard Mamba model without LayerNorm.**

In addition, we also conduct simulations using a pretrained standard Mamba architecture with LayerNorm in a practical setting on WikiText103. In Figure 8 in Appendix D.4 in the revised version, we visualize the evolution of token norms across the blocks. Figure 8 shows that the tokens' dynamics in a full Mamba setting can be quite complicated. When the input-output projection matrix has only negative eigenvalues, the tokens do not necessarily tend toward zero. This situation explains the effect of LayerNorm on the tokens' dynamic. Hence, **with LayerNorm in the Mamba architecture, our theoretical results on the tokens' dynamics might no longer hold.**

**(2) Regarding the effects of practical components of Mamba in the token reordering technique:** We conducted an ablation study on WikiText103 and examined the following configurations, both with and without the reordering technique:

(i) the full Mamba model,

(ii) the Mamba with weight sharing across layers,

(iii) the Mamba without layer normalization,

(iv) the Mamba without the short convolution layer.

To ensure fairness, all configurations are standardized to approximately 129M parameters. The results which are shown in Table 4 below highlighting the contribution of each component to the model's performance under our reordering technique. Our reordering method consistently yields at least 1.49 PPL improvement in all settings, which is significant for language modeling task with WikiText103.

*Table 4: Ablation study on language modeling task with WikiText103.*
| Model | With reordering | Without reordering | Improvement with reordering |
|-|-|-|-|
| Mamba| **14.97**| 16.46| 1.49|
| Mamba + weight sharing| 17.77           | 20.51             | 2.74       |
| Mamba without LayerNorm      | 17.82           | 19.31             | 1.49       |
| Mamba without Convolution    | 21.81           | 23.38             | 1.57       |


We have added these results to Appendix D.5 of the revised paper.


**(3) Regarding the possibility of extending the analysis to the bidirectional case:** Our theoretical results on the dynamic behaviors of tokens may no longer hold for bidirectional SSMs (see [Zhu2024] for a definition of bidirectional SSMs). In a bidirectional SSM, input tokens are reordered in two distinct sequences: one in the forward order and the other in the backward order. These two sequences, containing the same tokens but in different arrangements, are processed separately by the SSM before being combined into a single sequence using a summation operator. According to our theoretical results, tokens in earlier positions tend to diverge at a slower rate under the divergence scenario. Consequently, the summation operator mixes token dynamics that diverge at different rates, leading to more complex overall behaviors. We believe that a comprehensive investigation of token dynamics in this bidirectional setting is an interesting direction for future research.

[Zhu2024] Zhu, Lianghui, et al. "Vision mamba: Efficient visual representation learning with bidirectional state space model." arXiv preprint arXiv:2401.09417 (2024).

We would be happy to do any follow-up discussion or address any additional comments.

---

### Author Response · Authors · 2024-11-24
**General Responses 3 - An additional experiment**

Dear AC and Reviewers,

We would like to thank the reviewer again for your thoughtful reviews and valuable feedback. Beside of our previous responses, we would like to provide missing answer on the **effectiveness of our reordering technique across a broad range of architectures** below.

**Regarding the effectiveness of our reordering technique across a broad range of architectures:** We conducted an additional experiment to evaluate the effectiveness of our reordering technique across a broad range of architectures. We use the language modeling task on the WikiText-103 benchmark to test three models: Mamba, H3 [1], and Transformer [2]. For the latter two models, no direct equivalent of the step size matrix $S_\Delta$ exists for calculating the token importance score, as defined in Equation (14) of our paper. To adapt the reordering technique to these models, we made the following adjustments:

- **H3 Model**: The token importance score is computed as $s = \langle x, K \rangle$, where $K$ is a learnable vector.

- **Transformer Model**: We converted the pretrained GPT-2 small model into the Mamba format and fine-tuned it, following the method outlined in [3]. On the converted model, we calculated the token importance score as described in Equation (14).

The results, presented in Table 5, demonstrate that our reordering method consistently improves perplexity (PPL) across all tested models. These findings highlight the potential of our approach to enhance the performance of a wide range of architectures.


*Table 5: Test perplexity on different models*
|Name|With reordering| Without reordering|Improvement with reordering |
|-|-|-|-|
| Mamba | **14.97**| 16.46 |1.49|
| H3 | **20.72**|21.61 |0.89|
| Transformer |**13.44**|16.98|3.54|

We have added these results to Appendix D.6 of the revised paper.

We would be happy to do any follow-up discussion or address any additional comments.

**References**

[1] Fu, Daniel Y., et al. "Hungry hungry hippos: Towards language modeling with state space models." arXiv preprint arXiv:2212.14052 (2022).

[2] Vaswani, A. "Attention is all you need." Advances in Neural Information Processing Systems (2017).

[3] Wang, Junxiong, et al. "The mamba in the llama: Distilling and accelerating hybrid models." arXiv preprint arXiv:2408.15237 (2024).

---

### Meta-Review · Area_Chair_FoEg · 2024-12-26

**Metareview:**

This paper presents a theoretical analysis of token dynamics in deep selective ssms, particularly focusing on mamba. The key findings are:

- In the one-dimensional case, tokens either converges to zero or diverge to infinity, with specific criteria determining which scenario occurs;
- In the divergent case, tokens diverge at different rates based on their position, leading to unequal contributions during training.

authors propose two improvements: excluding scenarios where tokens converge to zero and reordering tokens based on importance scores. The authors validate these improvements through experiments on language modeling and image classification tasks.

Strengths:

- good theoretical analysis for ssms
- good empirical validation across multiple tasks and architectures

Weaknesses:

- Initial analysis relied on simplifying assumptions (1D case, time-independent parameters)
- Limited discussion of computational complexity and practical implementation challenges
- Initial experimental scope was narrow, focusing mainly on vision tasks

Authors thoroughly addressed reviewer concerns during rebuttal with additional experiments. I vote for acceptance.

**Additional Comments On Reviewer Discussion:**

in the discussion phase athors made the following changes that made all reviewers satisfied.

- tried to provide simulations showing theoretical findings for higher dimensions
- added language modeling (WikiText-103) and text classification tasks (IMDB, AGNews) (still small)
- provided detailed ablation studies
- conducted additional experiments showing behavior with/without LayerNorm and convolution layers.
- added comparisons with H3 and Transformer models.

---

### Decision · Program_Chairs · 2025-01-22

Accept (Spotlight)